# Inclusion of cGAMP within virus-like particle vaccines enhances their immunogenicity

Lise Chauveau[1,‡] , Anne Bridgeman[1,†], Tiong K Tan[1,†], Ryan Beveridge[2,3], Joe N Frost[1], Pramila Rijal[1], Isabela Pedroza-Pacheco[4], Thomas Partridge[4], Javier Gilbert-Jaramillo[5,6], Michael L Knight[5], Xu Liu[5,7], Rebecca A Russell[5], Persephone Borrow[4] , Hal Drakesmith[1], Alain R Townsend[1] & Jan Rehwinkel[1,*]

## Abstract

Cyclic GMP-AMP (cGAMP) is an immunostimulatory molecule produced by cGAS that activates STING. cGAMP is an adjuvant when administered alongside antigens. cGAMP is also incorporated into enveloped virus particles during budding. Here, we investigate whether inclusion of cGAMP within viral vaccine vectors enhances their immunogenicity. We immunise mice with virus-like particles (VLPs) containing HIV-1 Gag and the vesicular stomatitis virus envelope glycoprotein G (VSV-G). cGAMP loading of VLPs augments CD4 and CD8 T-cell responses. It also increases VLP- and VSV-G-specific antibody titres in a STING-dependent manner and enhances virus neutralisation, accompanied by increased numbers of T follicular helper cells. Vaccination with cGAMP-loaded VLPs containing haemagglutinin induces high titres of influenza A virus neutralising antibodies and confers protection upon virus challenge. This requires cGAMP inclusion within VLPs and is achieved at markedly reduced cGAMP doses. Similarly, cGAMP loading of VLPs containing the SARS-CoV-2 Spike protein enhances Spike-specific antibody titres. cGAMP-loaded VLPs are thus an attractive platform for vaccination.

**Keywords** cGAMP; influenza A virus; SARS-CoV-2; type I interferon; viral vaccine vector

**Subject Categories** Immunology; Methods & Resources; Microbiology, Virology & Host Pathogen Interaction

## Introduction

Vaccination is a powerful strategy in the fight against infectious diseases, including virus infection. Indeed, vaccination led to the global eradication of smallpox and is highly protective against some viruses including measles virus and yellow fever virus. However, the development of vaccines inducing long-lasting and broadly effective protection has been difficult for other viruses such as human immunodeficiency virus (HIV) and influenza A virus (IAV), highlighting the need for new vaccination strategies (Rappuoli *et al*, 2011).

Successful vaccines induce potent adaptive immune responses. Vaccine-mediated protection against most virus infections is thought to be predominantly due to induction of antiviral antibody responses that prevent or rapidly control subsequent infection. Antibody responses limit virus infection and spread through several mechanisms (Pelegrin *et al*, 2015). The most efficient one is mediated by neutralising antibodies that directly bind virus particles and prevent them from infecting cells. Following immunisation, antibodies are initially produced by short-lived extrafollicular plasmablasts. To achieve long-term protection, long-lived plasma cells and memory B cells must be generated in secondary lymphoid tissues. This process occurs in specialised structures called germinal centres (GCs) where B cells interact with multiple cell types that promote their maturation and differentiation into cells producing high-affinity antibodies that confer durable protection (Linterman & Hill, 2016; Cyster & Allen, 2019). CD4 T follicular helper (Tfh) cells are a CD4 T-cell subset specialised in providing help to B cells and are essential for GC formation. They increase the magnitude and quality of the humoral response by promoting B-cell proliferation, isotype switching and plasma cell differentiation; by mediating selection of high-affinity B cells in GCs; and by supporting the generation of

1 Medical Research Council Human Immunology Unit, Radcliffe Department of Medicine, Medical Research Council Weatherall Institute of Molecular Medicine, University of Oxford, Oxford, UK
2 MRC Molecular Hematology Unit, MRC Weatherall Institute of Molecular Medicine, John Radcliffe Hospital, University of Oxford, Oxford, UK
3 Virus Screening Facility, MRC Weatherall Institute of Molecular Medicine, John Radcliffe Hospital, University of Oxford, Oxford, UK
4 Nuffield Department of Clinical Medicine, University of Oxford, Oxford, UK
5 Sir William Dunn School of Pathology, University of Oxford, Oxford, UK
6 Department of Physiology, Anatomy and Genetics, University of Oxford, Oxford, UK
7 Key Laboratory of Human Disease Comparative Medicine, National Health Commission of China (NHC), Institute of Laboratory Animal Science, Peking Union Medicine College, Chinese Academy of Medical Sciences, Beijing, China
 *Corresponding author. Tel: +44 1865 222 362; E-mail: jan.rehwinkel@imm.ox.ac.uk
 †These authors contributed equally to this work
 ‡Present address: Institut de recherche en infectiologie de Montpellier (IRIM), CNRS UMR 9004, Montpellier, France

long-lived plasma cells and memory B cells (Crotty, 2019). In contrast, CD4 T follicular regulatory (Tfr) cells are involved in limiting GC reactions to prevent autoantibody formation. Therefore, the Tfh/Tfr ratio is important for regulation of GC responses (Sage *et al*, 2013).

Virus-specific cytotoxic T-cell (CTL) responses mediate clearance of infected cells to prevent virus spread and eradicate infection. If sterilising immunity is not conferred by antibodies, CTLs can make a key contribution to vaccine efficacy (Hansen *et al*, 2011). CTLs exert their activity by triggering destruction of infected cells via release of perforins and granzymes; by ligation of death-domain containing receptors and/or secretion of TNFα; and by producing "curative" cytokines such as IFNγ. Both the magnitude, i.e. the number of activated cells, and polyfunctionality of the T-cell response, i.e. the capacity to mediate a breadth of effector activities including production of multiple cytokines, are important determinants of CD8 T-cell-based vaccine efficacy (Panagioti *et al*, 2018).

Initiation of virus-specific CD4 and CD8 T-cell responses requires presentation of viral antigens to naïve T cells by professional antigen-presenting cells (APCs), principally dendritic cells (DCs). T cells need to receive three signals for activation: T-cell receptor (TCR) triggering by contact with peptide-major histocompatibility complexes (MHC) (signal 1), costimulatory signals (signal 2) and inflammatory cytokines (signal 3) (Joffre *et al*, 2009).

Therefore, to induce adaptive immune responses, vaccines need to contain not only an appropriate antigen but also an adjuvant. Adjuvants exert a breath of effects; for example, they induce the expression of costimulatory molecules and cytokines by DCs (Coffman *et al*, 2010). There are only a limited number of FDA-approved adjuvants, most of which are based on aluminium salts (Shi *et al*, 2019). The increasing knowledge in the field of innate immunity, particularly in the mechanisms underlying pathogen recognition by innate immune receptors, provides an opportunity to develop new adjuvants that specifically engage such receptors and trigger a robust response (Temizoz *et al*, 2018). Adjuvants targeting Toll-like receptors or the cytosolic DNA sensing pathway have attracted a lot of attention (Dubensky *et al*, 2013). In particular, cyclic dinucleotides (CDNs), which activate stimulator of interferon genes (STING, also known as TMEM173, MPYS, ERIS and MITA) and induce a type I interferon (IFN-I) response as well as production of pro-inflammatory cytokines, are being developed as adjuvants (Cai *et al*, 2014). CDNs facilitate both CD8 T-cell and antibody responses (Li *et al*, 2013; Blaauboer *et al*, 2014; Kuse *et al*, 2019) and are effective as mucosal adjuvants (Ebensen *et al*, 2011; Blaauboer *et al*, 2015). 2′3′ cyclic GMP-AMP (cGAMP) is of particular interest. It is produced by cGAMP synthase (cGAS) upon DNA sensing in the cell cytoplasm (Ablasser *et al*, 2013; Diner *et al*, 2013; Sun *et al*, 2013). Soluble cGAMP has been employed as an adjuvant in multiple pre-clinical vaccination models and is an anti-tumour agent (Li *et al*, 2013; Blaauboer *et al*, 2015; Corrales *et al*, 2015; Demaria *et al*, 2015; Temizoz *et al*, 2015; Lee *et al*, 2016; Li *et al*, 2016; Liu *et al*, 2016; Wang *et al*, 2016; Borriello *et al*, 2017; Takaki *et al*, 2017; Wang *et al*, 2017; Gutjahr *et al*, 2019; Luo *et al*, 2019; Vassilieva *et al*, 2019; Wang *et al*, 2020). However, soluble cGAMP levels are likely to diminish quickly in the extracellular milieu, due to diffusion from the site of administration and degradation by phosphodiesterases such as ectonucleotide pyrophosphatase/phosphodiesterase 1 (ENPP1), an enzyme degrading extracellular

ATP and cGAMP (Li *et al*, 2014; Carozza *et al*, 2020). Indeed, when injected intra-muscularly, the concentration of cGAMP at the inoculation site decreases rapidly, resulting in a sub-optimal adjuvant effect (Wang *et al*, 2016).

We and others previously showed that cGAMP is packaged into nascent viral particles as they bud from the membrane of an infected cell (Bridgeman *et al*, 2015; Gentili *et al*, 2015). Upon virus entry into newly infected cells, cGAMP is released into the cytosol and directly activates STING. Here, we took advantage of this observation and hypothesised that inclusion of the adjuvant cGAMP in viral vaccine vectors may enhance their immunogenicity by targeting adjuvant and antigen to the same cell and by protecting cGAMP from degradation in the extracellular environment. Indeed, using HIV-derived virus-like particles (VLPs), we found that the presence of cGAMP within VLPs enhanced adaptive immune responses to VLP antigens. Antigen-specific CD4 and CD8 T-cell responses were augmented, as well as neutralising antibody production. The latter was accompanied by an increase in Tfh cells in draining lymph nodes. The increased production of VLP-specific antibodies required STING. cGAMP-loaded VLPs containing the IAV haemagglutinin protein induced neutralising antibodies and conferred protection against development of severe disease after challenge with live IAV. Moreover, cGAMP-loaded VLPs provided protection at low doses, which is advantageous for vaccine production, and could improve safety. Finally, Spike-specific antibody titres were increased when we included cGAMP within VLPs containing the SARS-CoV-2 Spike protein, demonstrating the versatility of cGAMP-loaded VLPs for immunisation. Taken together, our proof-of concept study highlights the utility of cGAMP-loaded VLPs as a vaccine platform.

# Results

## cGAMP loading of HIV-derived VLPs

HIV-derived viral vectors and VLPs are routinely produced in the cell line HEK293T by transfection of plasmids encoding viral components (Milone & O'Doherty, 2018). Here, we generated VLPs by using plasmids expressing the HIV-1 capsid protein Gag fused to GFP (Gag-GFP) and the vesicular stomatitis virus envelope glycoprotein G (VSV-G). The resulting VLPs consist of a Gag-GFP core and a lipid membrane derived from the producer cell that is spiked with VSV-G proteins. Of note, these VLPs do not contain viral nucleic acid and can therefore not replicate in the host (Deml *et al*, 2005). Additional over-expression of cGAS in the VLP producer cells results in its activation, presumably by the transfected plasmid DNA, and in the presence of cGAMP in the cytosol. It is noteworthy that HEK293T cells do not express STING (Burdette *et al*, 2011); therefore, cGAS-overexpressing VLP producer cells do not respond to the presence of cGAMP. cGAMP is then packaged into the nascent viral particles, which are released as cGAMP-loaded VLPs (hereafter cGAMP-VLPs; Fig 1A). As a control, we produced VLPs that do not contain cGAMP (Empty-VLPs) by using a catalytically inactive version of cGAS.

To assess the efficiency of cGAMP incorporation into our VLPs, we extracted small molecules from VLPs as previously described (Mayer *et al*, 2017). cGAMP in the extract was then quantified by ELISA. While Empty-VLPs did not contain detectable levels of

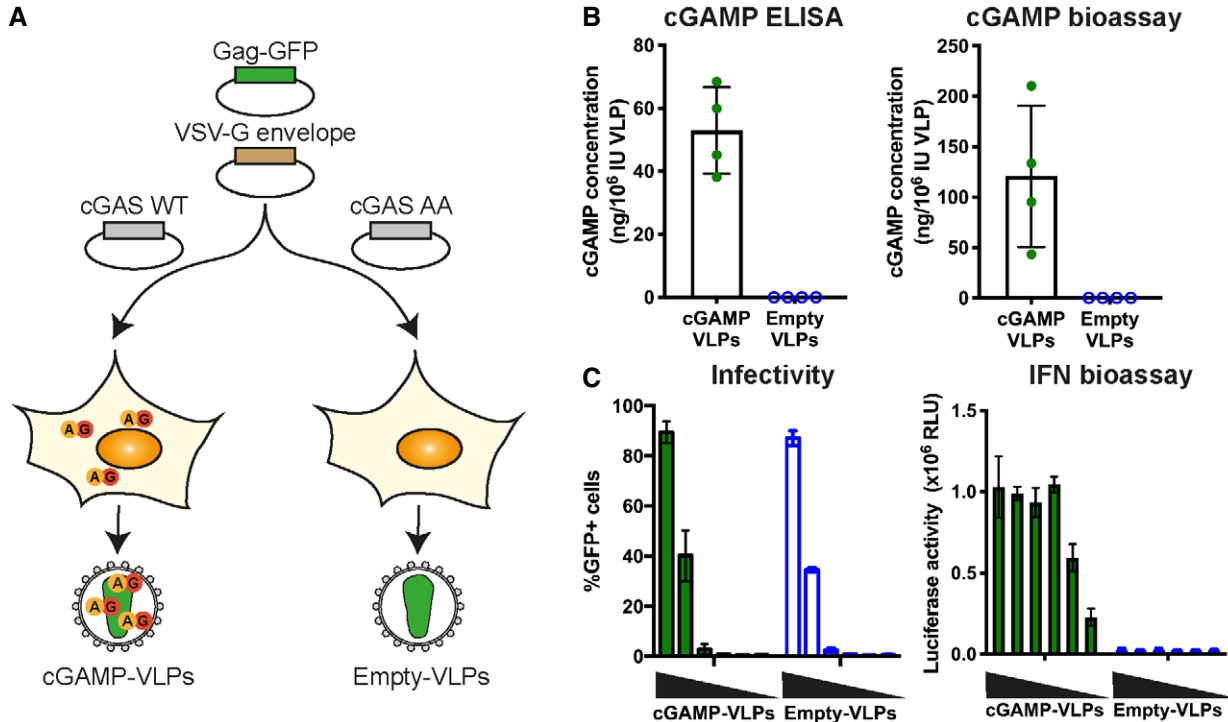

**Figure 1. cGAMP incorporated into Gag-GFP virus-like particles (VLPs) induces IFN-I in infected cells.**

A  Schematic representation of cGAMP- and Empty-VLP production. HEK293T cells were transfected with plasmids encoding HIV-1 Gag-GFP and VSV-G envelope to enable VLP production. Overexpression of cGAS WT in the same cells generated cGAMP that was then incorporated into nascent VLPs (cGAMP-VLPs). As control, Empty-VLPs were produced in cells where a catalytically inactive cGAS (cGAS AA) was overexpressed.

B  cGAMP is incorporated into cGAMP-VLPs. Small molecules were extracted from VLP preparations, and the cGAMP concentration was measured using a cGAMP ELISA and a cGAMP bioassay.

C  cGAMP-VLPs induce an IFN-I response in target cells. HEK293 cells were infected with decreasing amounts of cGAMP-VLPs and Empty-VLPs (1/5 serial dilutions starting at 2 µl of VLP stocks per well), and the infection was monitored 24 h later by quantifying GFP$^+$ cells by flow cytometry. Supernatants from the same infected cells were then transferred to a reporter cell line expressing firefly luciferase under a promoter induced by IFN-I (ISRE). Luciferase activity measured 24 h later indicated the presence of IFN-I in the supernatants.

Data information: Data in (B) are pooled from four independent VLP productions. Each symbol corresponds to one VLP production, and mean and SD are shown. Data in (C) are pooled from three independent VLP productions tested simultaneously in technical duplicates in infectivity and IFN-I bioassays; mean and SD are shown.

cGAMP, cGAMP-VLPs contained on average ~50 ng cGAMP per $10^6$ infectious units (IU) of VLPs (Fig 1B). To confirm this result, we used a cGAMP bioassay in which semi-permeabilised THP-1 cells are stimulated with extracts from VLPs as described previously (Bridgeman et al, 2015). cGAMP levels determined by this assay were 2- to 3-fold higher compared with the ELISA data, with an average of ~120 ng cGAMP per $10^6$ IUs (Fig 1B). These results were within the same range. The bioassay was likely less accurate as it involved stimulation of cells, and we therefore based further experiments on the ELISA results. Importantly, Empty-VLPs did not contain detectable levels of cGAMP measured by either method. cGAMP-VLPs and Empty-VLPs were equally infectious in HEK293 cells (Fig 1C). To confirm that cGAMP-VLPs trigger an IFN-I response, supernatant from the STING-positive HEK293 cells used in the infectivity assay was transferred to a reporter cell line expressing firefly luciferase under the interferon-sensitive response element (ISRE) promoter (Bridgeman et al, 2015). At similar infection rates, cGAMP-VLPs induced IFN-I production while Empty-VLPs did not (Fig 1C). Taken together, these results show that

cGAMP can be efficiently packaged into VLPs consisting of HIV-1 Gag-GFP and the VSV-G envelope.

## Immunisation with cGAMP-VLPs induces higher and more polyfunctional CD4 and CD8 T-cell responses compared with Empty-VLPs

To test whether cGAMP-VLPs induce a better immune response than Empty-VLPs *in vivo*, we injected C57BL/6 mice intra-muscularly with $10^6$ IU of cGAMP-VLPs or Empty-VLPs or, as a control, PBS. We first assessed CD4 T-cell responses in the spleen 14 days after immunisation. As we were unable to identify a specific peptide epitope within HIV-Gag recognised by CD4 T cells in H-2$^b$ mice, we used bone marrow-derived myeloid cells (BMMCs) pulsed with cGAMP-VLPs for the evaluation of antigen-specific CD4 T-cell responses. We co-cultured these cells with splenocytes for 6 h before assessing IL2, IFNγ and TNFα production by CD4 T cells by intracellular cytokine staining (ICS). Compared to mice immunised with Empty-VLPs, we observed 2.7-fold increased frequencies of

CD4 T cells producing each of these cytokines in response to VLP-pulsed BMMCs in mice immunised with cGAMP-VLPs (Fig 2A and B). Moreover, cGAMP enhanced the proportion of cells that were able to co-produce two or all three cytokines (Fig 2C; 2.1- and 3.7-fold increases, respectively). Similar results were obtained when BMMCs were pulsed with Empty-VLPs (Fig EV1A and B).

We next assessed CD8 T-cell responses following immunisation. We screened a panel of overlapping 15-mer peptides spanning the HIV-1 Gag sequence and identified a peptide that stimulated an IFNγ response in cells from spleen in IFNγ ELISPOT assays (peptide p92; Fig EV1C and D). We then used NetMHCpan-3.0 (http://www.cbs.d tu.dk/services/NetMHCpan-3.0/; Nielsen & Andreatta, 2016) to predict the optimal epitope sequence recognised within p92 and identified a 9-mer peptide (SQVTNSATI, termed HIV-SQV) that triggered T-cell recognition more efficiently than the original 15-mer peptide (Fig EV1E and F). This 9-mer peptide was also reported to constitute an immunodominant HIV-1 Gag epitope in H-2[b] mice in a prior study (Holechek et al, 2016). The HIV-SQV peptide was used for all subsequent analyses of VLP-elicited CD8 T-cell responses. We evaluated responses to the HIV-SQV peptide by IFNγ ELISPOT assay and showed that, when compared to Empty-VLPs, cGAMP-VLPs induced a modest but significant increase in the magnitude of the response (Fig 2D, 1.7-fold increase). To assess whether cGAMP loading of VLPs also enhanced the polyfunctionality of the responding CD8 T cells, we stimulated splenocytes for 6 h and stained for upregulation of CD107a (LAMP-1), a degranulation marker identifying cytotoxic cells, and for the production of IFNγ, TNFα and IL2 by ICS. Paralleling the results from the ELISPOT assay, CD8 T cells from mice immunised with cGAMP-VLPs showed a modest but significant increase in the frequency of cells upregulating CD107a (1.6-fold increase) and/or producing IFNγ (2-fold) and/or TNFα (1.9-fold) (Fig 2E and F). Furthermore, cGAMP enhanced the proportion of CD8 T cells that were able to co-produce two of the cytokines evaluated (Fig 2G; 1.9-fold increase).

Control of vaccinia virus infection by the immune system relies in part on CD8 T-cell responses (Xu et al, 2004). As immunisation with cGAMP-VLPs increased anti-HIV-Gag CD8 T-cell responses, we assessed whether this resulted in increased protection against subsequent infection with a vaccinia virus expressing the same HIV-Gag (vVK1 (Karacostas et al, 1989)). One month after immunisation, mice were challenged with vVK1, and 5 days after infection virus load in the ovaries was assessed by plaque assay. We observed no weight loss over the course of the infection (Fig EV2A). Immunisation with both VLPs reduced vVK1 load, and cGAMP-VLP-immunised mice showed a slight but non-significant increase in protection compared with animals immunised with Empty-VLPs (Fig EV2B and C).

Taken together, these results demonstrate that cGAMP loading of VLPs enhances polyfunctional CD4 and CD8 T-cell responses to VLP antigens.

## cGAMP loading of VLPs enhances STING-dependent serum titres of VLP binding and neutralising antibodies

Next, we assessed the antibody response in immunised mice. We set up ELISAs that allow detection of serum antibodies binding to any protein in the VLPs (using lysates from cGAMP-VLPs for coating), or of antibodies specific for the VSV-G envelope or the

HIV-Gag protein. In mice immunised with VLPs, we detected very strong IgG responses and lower-titre IgM responses targeting VLP proteins 14 days after immunisation, indicating antibody class-switching (Figs 3A and EV3A). Interestingly, immunisation with cGAMP-VLPs induced stronger anti-VLP antibody responses compared with the Empty-VLP-immunised group, with statistically significant differences being observed in IgG2a/c and IgG2b levels. We also detected IgG antibodies targeting the VSV-G envelope, and IgG2a/c and IgG2b titres were significantly higher in the cGAMP-VLP-immunised group (Figs 3B and EV3A). Titres of antibodies recognising the intracellular antigen HIV-Gag were low or undetectable, but a similar trend was observed for a higher-magnitude response in the cGAMP-VLP-immunised group (Fig EV3B). Increased IgG2b anti-VLP antibody titres were also observed in cGAMP-VLP-immunised mice when sera were tested by ELISA using lysates from Empty-VLPs for coating (Fig EV3C).

We next tested whether the antibody response induced by cGAMP-VLPs was dependent on STING signalling. We immunised WT- or STING-deficient mice (Tmem173−/−; hereafter referred to as Sting−/− for simplicity) with VLPs and assessed the anti-VLP IgG2b response at day 14 after immunisation (Fig 3C). As a control for an adjuvant that does not depend on STING, we supplemented Empty-VLPs with 25 μg poly(I:C), an immunostimulatory RNA that signals through Toll-like receptor 3 and RIG-I-like receptors (Empty-VLPs + poly(I:C)). In WT mice, immunisation with cGAMP-VLPs and Empty-VLPs + poly(I:C) each induced high titres of IgG2b antibodies that exceeded those elicited in the Empty-VLP group (Fig 3C). In Sting−/− mice, immunisation with cGAMP-VLPs induced lower levels of IgG2b antibodies that were comparable with those in the Empty-VLP group, while the response to immunisation with Empty-VLPs + poly(I:C) remained unaffected (Fig 3C).

To test whether the anti-VLP antibodies were neutralising, we assessed the in vitro neutralisation capacity of sera using a VSV-G pseudotyped HIV-1-based lentivector expressing GFP (Fig 3D). The effect of pre-incubation with serum samples on the infectivity of the HIV-1-GFP virus was measured by monitoring GFP expression in HEK293 cells (Fig EV3D and E). Although immunisation with both cGAMP-VLPs and Empty-VLPs induced neutralising antibodies, this response was stronger when cGAMP was present within the VLPs, and sera from cGAMP-VLP immunised mice showed a 2.5 times higher half-maximal inhibitory concentration (Fig 3E and F).

In summary, immunisation with cGAMP-VLPs induced an increased antibody response that targeted proteins from total VLP lysates including the VSV-G envelope protein, and this anti-VLP response was dependent on STING signalling. Moreover, cGAMP loading enhanced production of virus neutralising antibodies.

## Incorporation of cGAMP into VLPs increases the CD4 Tfh cell response

To gain insight into how immunisation with cGAMP-VLPs resulted in an increased antibody response, we investigated B- and T-cell populations in inguinal lymph nodes that drain the injection site. As CD4 T-cell responses were increased in the spleens of cGAMP-VLP-immunised mice, we first tested whether follicular CD4 T-cell numbers were elevated in lymphoid tissues draining the immunisation site. We identified follicular CD4 T cells as

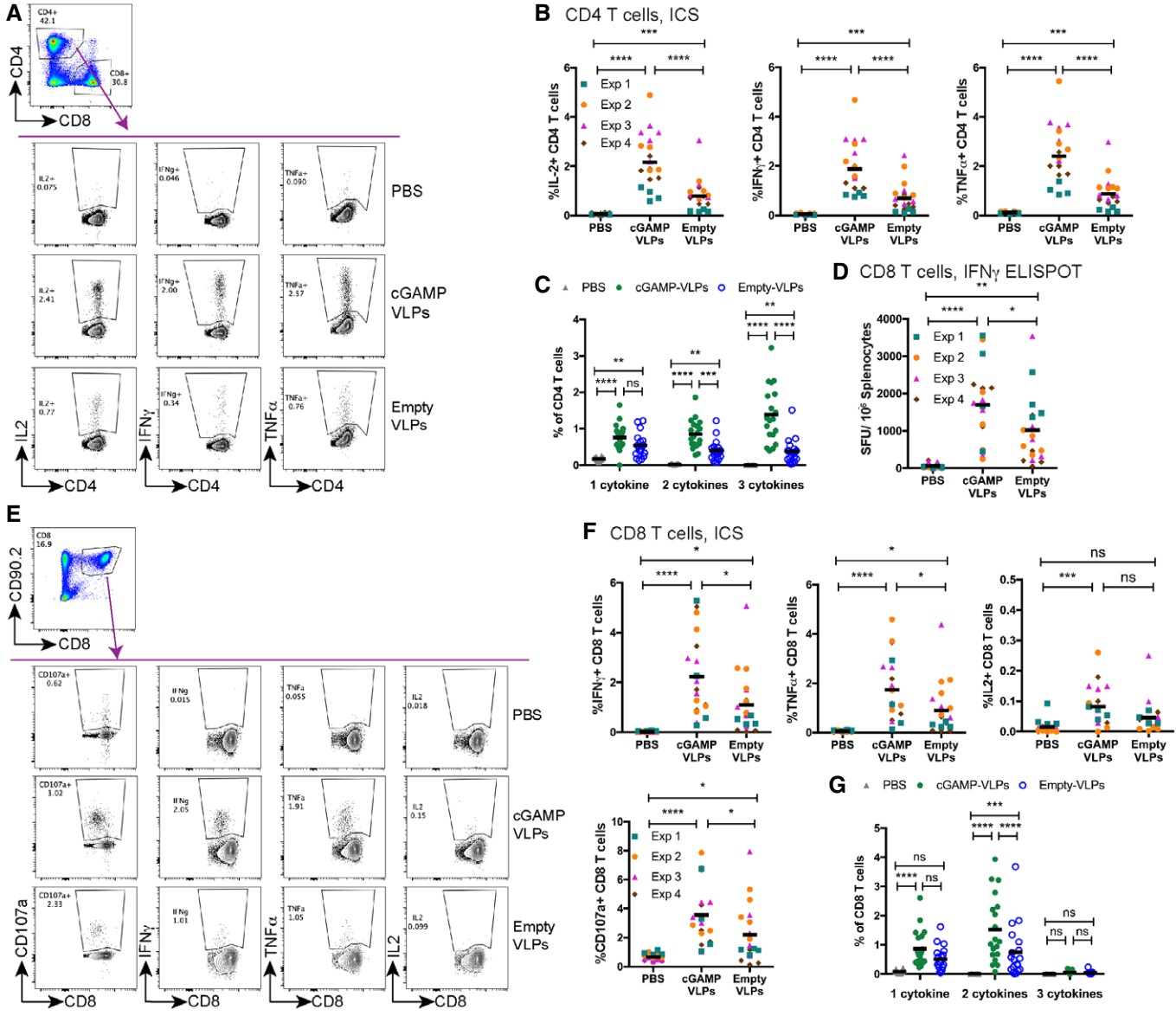

**Figure 2. cGAMP loading of VLPs increases the magnitude of the CD4 and CD8 T-cell responses elicited after immunisation.**

C57BL/6 mice were injected with cGAMP-VLPs, Empty-VLPs or PBS as a control *via* the intra-muscular route. 14 days later, VLP-specific T-cell responses were evaluated in the spleen.

A–C   Immunisation with cGAMP-VLPs enhances VLP-specific CD4 T-cell responses. BMMCs from C57BL/6 mice were pulsed overnight with cGAMP-VLPs and used to stimulate cells from spleens of immunised mice. Cells were co-cultured for 6 h prior to evaluation of CD4 T-cell responses by ICS. CD4 T cells were gated as live, MHC-II[−], CD4[+], CD8[−]. CD4 T cells expressing IL2, IFNγ or TNFα were analysed as shown in (A). The percentage of total CD4 T cells producing each cytokine is shown in (B), and the percentage of CD4 T cells co-producing 1, 2 or 3 cytokines is shown in (C).

D–G   Immunisation with cGAMP-VLPs facilitates induction of HIV-1 Gag-specific polyfunctional CD8 T-cell responses. Cells from spleens of immunised mice were stimulated with the HIV-SQV peptide. IFNγ-producing cells were enumerated by ELISPOT 24 h after stimulation with peptide (D). Alternatively, cells were analysed by ICS 6 h after stimulation with peptide. CD8 T cells were gated as live, CD90.2[+], CD8[+]. CD8 T cells expressing CD107a, IFNγ, TNFα or IL2 were analysed as shown in (E). Panel F shows the percentage of total CD8 T cells upregulating CD107a and/or producing each cytokine, and panel G shows the percentage of CD8 T cells co-producing 1, 2 or 3 cytokines.

Data information: Panels (A) and (E) show representative examples of data from four independent experiments. In panels (B–D), (F) and (G), data are pooled from four independent experiments. A total of 19 mice was analysed per condition. Symbols show data from individual animals, and in (B), (D) and (F) are colour-coded by experiment. Horizontal lines indicate the mean. Statistical analyses were performed using a 2-way ANOVA followed by Tukey's multiple comparisons test. In (B), (D) and (F) data were blocked on experiments. ns $P \geq 0.05$; *$P < 0.05$; **$P < 0.01$; ***$P < 0.001$; ****$P < 0.0001$. See also Figs EV1 and EV2.

CD4[+]CD44[+]PD1[hi]CXCR5[hi] and subdivided them into Tfh and Tfr cells by analysing FoxP3, which is expressed in Tfr cells (Fig 4A). Immunisation with VLPs led to an increase in the proportion of

follicular T cells within the CD4 T-cell population in the draining lymph node (Fig 4A and B). This was due to an expansion of Tfh cells, as the latter increased significantly in frequency after VLP

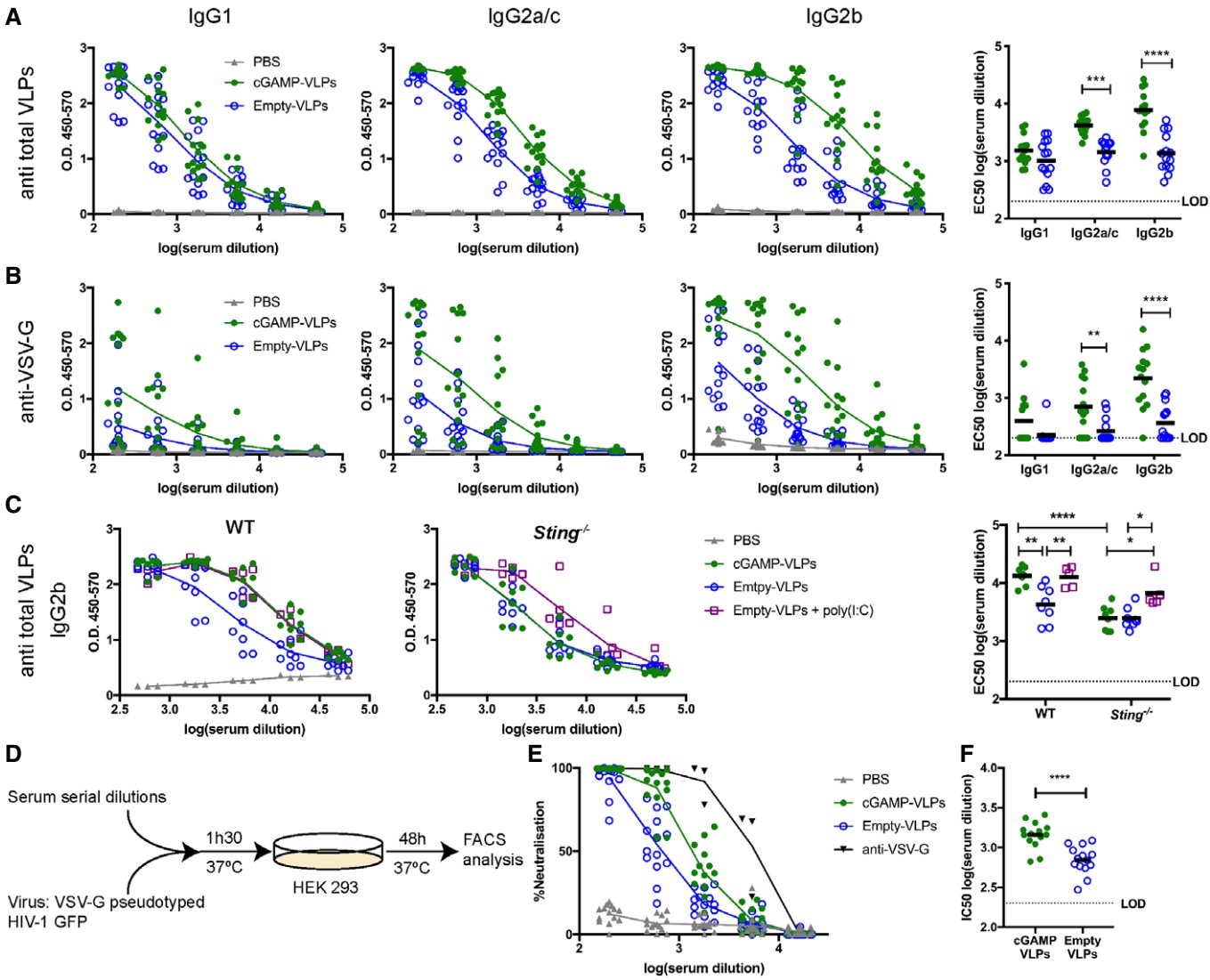

**Figure 3. Immunisation with VLPs containing cGAMP increases neutralising antibody responses.**

C57BL/6 mice were injected with cGAMP-VLPs, Empty-VLPs or PBS as a control *via* the intra-muscular route. Serum antibody responses were evaluated 14 days later.

A, B   cGAMP loading enhances IgG responses specific to VLP proteins, including VSV-G. ELISA plates were coated with lysate from cGAMP-VLPs (A) or recombinant VSV-G protein (B). Antibodies of different isotypes specific for these proteins were measured in sera from immunised mice by ELISA. The optical density at increasing serum dilutions is shown in the first three graphs from the left, and the EC50 is on the right.

C   Enhanced antibody production following immunisation with cGAMP-VLPs relies on STING signalling. WT or $Sting^{-/-}$ mice were immunised with PBS, cGAMP-VLPs, Empty-VLPs or Empty-VLPs + poly(I:C). IgG2b antibodies recognising VLP proteins were assessed by ELISA. The optical density at increasing serum dilutions is shown in the first two graphs from the left, and the EC50 is on the right.

D–F   Immunisation with cGAMP-VLPs enhances production of neutralising antibodies. Serial dilutions of serum samples from individual mice were incubated with VSV-G pseudotyped HIV-1-GFP for 90 min at 37°C before infection of HEK293 cells. As a control, serial dilutions of the anti-VSV-G neutralising antibody 8G5F11 were tested in parallel. After 2 days, infection was measured by quantifying GFP+ cells by flow cytometry (D). Neutralising capacities of serum samples from individual animals were calculated as a percentage of neutralisation (calculated relative to the maximum infection in each experiment) (E) and as the half-maximal inhibitory concentration (IC50) (F).

Data information: In (A, B, E and F), data are pooled from three independent experiments and a total of 14 mice was analysed per condition. In (C), data are pooled from two independent experiments and 5–8 mice were analysed per condition. Symbols show data from individual animals, and the means are indicated. Statistical analyses were done using a 2-way ANOVA followed by Tukey's multiple comparisons test (A–C) or a Kruskal–Wallis test followed by Dunn's multiple comparisons test (F). ns $P \geq 0.05$; *$P < 0.05$; **$P < 0.01$; ***$P < 0.001$; ****$P < 0.0001$. See also Fig EV3.

immunisation, whereas Tfr frequencies within CD4 T cells remained unaltered. As a consequence of this, there was a profound shift in the Tfh:Tfr ratio in VLP immunised as compared to control mice (Fig 4C). Importantly, the increase in Tfh cells was more

pronounced in cGAMP-VLP-immunised mice compared with Empty-VLP-injected animals (Fig 4B and C; 1.6-fold increase).

To assess the impact of this increased Tfh response on B-cell responses, we first gated on GC B cells (B220+IgD−CD95+GL7+ cells;

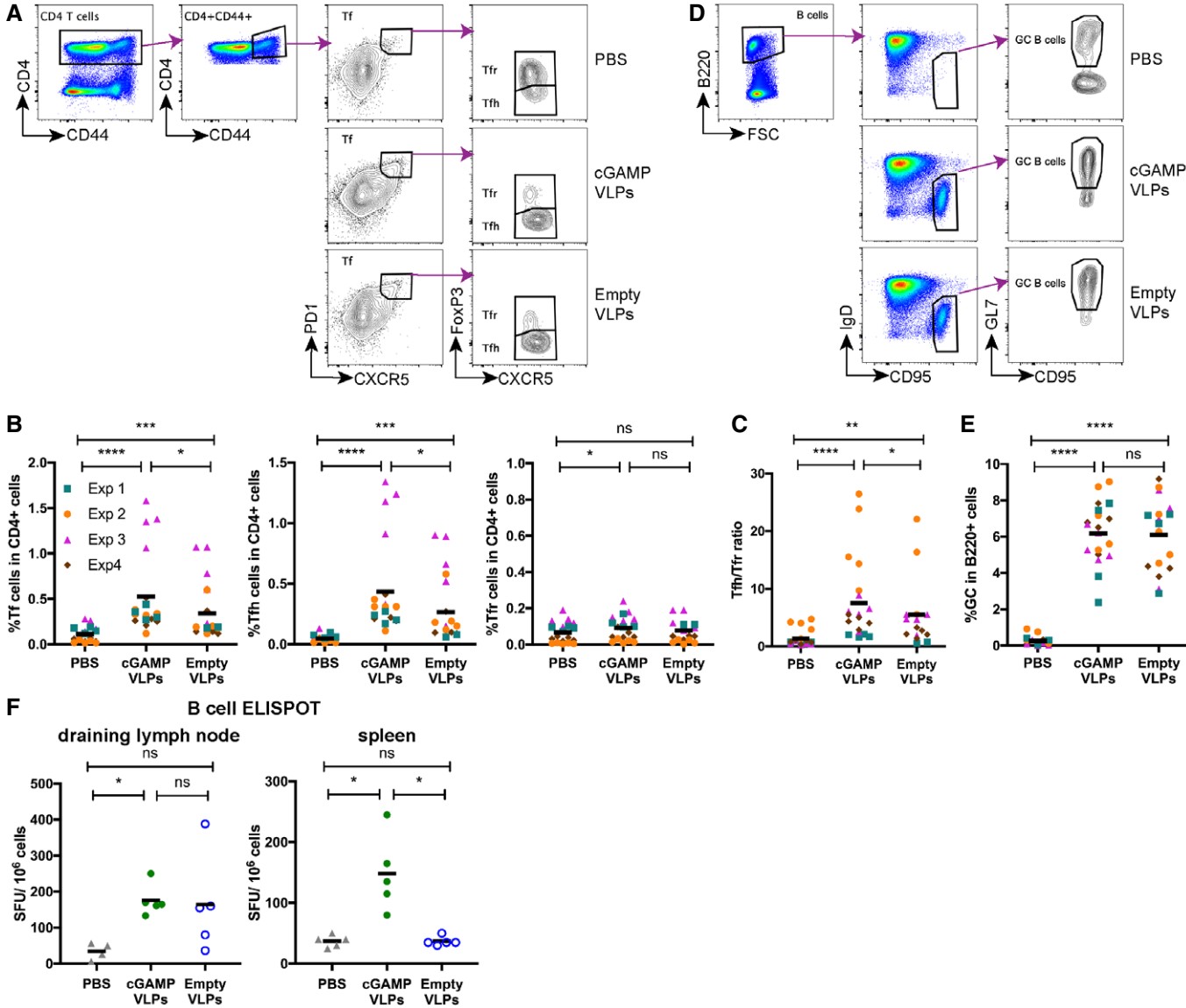

**Figure 4. cGAMP loading of VLPs enhances induction of CD4 Tfh responses.**

C57BL/6 mice were injected with cGAMP-VLPs, Empty-VLPs or PBS as a control *via* the intra-muscular route. 14 days later, T and B cells in the draining inguinal lymph nodes were characterised by flow cytometry and B-cell ELISPOT assays.

A–C  Immunisation with cGAMP-VLPs enhances accumulation of Tfh cells in the draining lymph node. T follicular (Tf) cells were identified by flow cytometry as CD4$^+$CD44$^+$CXCR5$^{hi}$PD1$^{hi}$ cells and were further subdivided into Tfr cells (FoxP3$^+$) and Tfh cells (FoxP3$^-$). The gating strategy is shown in (A), and the percentages of Tf, Tfh and Tfr cells within CD4$^+$ cells are shown in (B). The ratio of Tfh/Tfr is shown in (C).

D, E  Immunisation with VLPs induces germinal centre formation. Germinal centre B cells were identified by flow cytometry as B220$^+$IgD$^-$CD95$^+$GL7$^+$ cells. The gating strategy is shown in (D), and the percentage of germinal centre B cells amongst B220$^+$ cells is shown in (E).

F  Immunisation with cGAMP-VLPs increases production of antibody-secreting cells. Cells from draining lymph nodes and spleens were seeded in ELISPOT plates coated with cGAMP-VLP lysates. After overnight incubation, cells producing VLP-specific IgG antibodies were identified using an anti-IgG Fc antibody.

Data information: Panels (A) and (D) show representative examples of data from four independent experiments. In (B), (C) and (E), data were pooled from four independent experiments including a total of 19 mice analysed per condition. Symbols show data from individual animals and are colour-coded by experiment. In (F), symbols show data from 5 mice per group measured in duplicate in one experiment. Horizontal lines indicate the mean. Statistical analyses were done using a 2-way ANOVA followed by Tukey's multiple comparisons test (B, C, E) or a Kruskal–Wallis test followed by Dunn's multiple comparisons test (F). ns $P \geq 0.05$; *$P < 0.05$; **$P < 0.01$; ***$P < 0.001$; ****$P < 0.0001$.

Fig 4D). Immunisation with VLPs induced a robust GC B-cell response, with no difference being observed in the frequencies of GC B cells in cGAMP-VLP and Empty-VLP groups at the day 14 time-point analysed (Fig 4E). We next evaluated the generation of

antibody-secreting cells (ASCs) by antigen-specific B-cell ELISPOT assay on cells from both draining lymph nodes and spleens 14 days after immunisation. VLP-specific ASCs were detected in the lymph nodes of mice injected with both cGAMP-VLPs and Empty-VLPs

(Fig 4F). In the spleen, VLP-specific ASCs were also observed in cGAMP-VLP-immunised animals, but not in Empty-VLP-immunised animals (Fig 4F).

Taken together, these results suggest that immunisation with cGAMP-VLPs increased the antibody response by enhancing the accumulation of Tfh cells in draining lymph nodes, thereby promoting the development of ASCs.

## cGAMP-VLPs pseudotyped with IAV haemagglutinin induce a neutralising antibody response and confer protection following live virus challenge

As immunisation with cGAMP-VLPs induced high titres of neutralising antibodies, we explored whether they could confer protection following a live virus challenge. Protection against IAV infection correlates with serum antibodies that have haemagglutination inhibition activity (Krammer, 2019). We therefore produced cGAMP-VLPs and Empty-VLPs incorporating the IAV surface glycoprotein haemagglutinin (HA) from the mouse-adapted PR8 strain of IAV (designated cGAMP-HA-VLPs and Empty-HA-VLPs, respectively) (Fig EV4A). cGAMP-HA-VLPs and Empty-HA-VLPs were equally infective, as assessed by the percentage of GFP-positive cells observed following infection of HEK293 cells with titrated doses of VLPs (Fig EV4B). Staining of infected HEK293 cells with the 21-D8-5A monoclonal antibody recognising HA revealed the presence of similar percentages of HA$^+$ cells after infection with cGAMP-HA-VLPs and Empty-HA-VLPs (Fig EV4B). As control, cells infected with cGAMP-VLPs without HA showed no detectable staining. These data confirmed that HA was transferred by HA-VLPs to infected cells. Finally, we verified that supernatant from cells infected with cGAMP-HA-VLPs contained IFN-I, suggesting that the presence of HA did not affect the incorporation of cGAMP into the VLPs (Fig EV4C).

Next, we immunised mice with HA-VLPs. Two and 3 weeks after immunisation, sera were analysed for neutralising antibodies using a microneutralisation assay. In brief, a single cycle IAV expressing eGFP and PR8 HA was pre-incubated with sera and its infectivity was then monitored using MDCK-SIAT1 cells (Powell et al, 2012). Immunisation with both Empty-HA-VLPs and cGAMP-HA-VLPs induced neutralising antibodies, and the presence of cGAMP in the VLPs increased this response by 2.7-fold at week 2 and 2.5-fold at week 3 (Fig 5A and B). To determine whether immunisation conferred protection upon in vivo challenge with live IAV, we infected mice with $10^4$ TCID$_{50}$ (Median Tissue Culture Infectious Dose) of HA-matched PR8 IAV 1 month after immunisation. Animals immunised with $10^6$ infectious units of both VLPs were protected against the weight loss observed between days 3 and 4 after IAV infection in PBS-treated mice, both resulting in 100% survival (Fig 5C and D). This prompted us to reduce the amount of VLPs used for immunisation. At an intermediate dose of $2 \times 10^5$ infectious units of VLPs, cGAMP-HA-VLPs induced a 2.4-fold higher antibody response at week 3 (Fig 5B) and were fully protective against weight loss and disease progression to an end-point where humane sacrifice was necessary, while immunisation with Empty-HA-VLPs only delayed disease progression by about 4 days, resulting in 100 and 16.7% survival, respectively (Fig 5C and D). At the lowest dose of VLPs tested ($5 \times 10^4$ infectious units), cGAMP-HA-VLPs induced a 3.6-fold higher antibody response at week 3

(Fig 5B). Empty-HA-VLPs were not protective whereas cGAMP-HA-VLPs protected most animals against severe disease (83% survival) (Fig 5C and D).

To compare our strategy of loading VLPs with cGAMP with an unrelated adjuvant as a benchmark, we supplemented Empty-HA-VLPs with AddaVax, a squalene-based oil-in-water emulsion similar to the adjuvant currently approved in Europe for the influenza vaccine (MF59) (Ott et al, 1995). We found that both cGAMP-HA-VLPs and Empty-HA-VLPs injected together with AddaVax conferred full protection against subsequent IAV challenge and induced similar neutralising antibody titres (Fig 6).

To test whether the protective effect of cGAMP-loaded VLPs required cGAMP to be present within VLPs, we compared cGAMP-HA-VLPs with Empty-HA-VLPs mixed prior to immunisation with chemically synthesised cGAMP. In these experiments, we used $5 \times 10^4$ infectious units of cGAMP-HA-VLPs containing ~3.5 ng cGAMP (cGAMP content of cGAMP-HA-VLPs used was quantified by ELISA as in Fig 1B). We therefore administered either 3.5 ng or 35 ng cGAMP mixed with Empty-HA-VLPs (Empty-HA-VLPs + matched cGAMP or 10× cGAMP) to mice for comparison with cGAMP-HA-VLPs. One month after immunisation, mice were challenged with $10^4$ TCID$_{50}$ of HA-matched PR8 IAV. With the exception of a single animal receiving 10x cGAMP, Empty-HA-VLPs without or with added cGAMP failed to protect mice from weight loss whereas cGAMP-HA-VLPs were fully protective, as observed before (Fig 6A and B). These results were paralleled by serum neutralising antibody titres at week 3 (Fig 6C and D) and showed that, to exert a protective effect, cGAMP needed to be present within VLPs.

Taken together, these results show that vaccination with VLPs incorporating IAV HA induced neutralising antibodies in mice, which were protected against subsequent IAV challenge. Incorporation of cGAMP into HA-VLPs was as efficient as a benchmark adjuvant (AddaVax) in inducing protection against IAV. Moreover, the presence of cGAMP in HA-VLPs, but not exogenous cGAMP mixed with HA-VLPs, enhanced the neutralising antibody response and, particularly at lower doses of VLPs used for immunisation, facilitated protection against IAV.

## cGAMP loading of VLPs containing the SARS-CoV-2 Spike protein augments antibody responses

Finally, we wished to explore the versatility of cGAMP-loaded VLPs as vaccine vectors by incorporating another viral antigen. Given the urgent need for vaccines against SARS-CoV-2, we produced VLPs in HEK293T cells expressing the SARS-CoV-2 Spike (S) protein, a viral envelope protein that contains key antigens recognised by neutralising antibodies (Zhou et al, 2020). S is processed by cellular proteases into S1 and S2 subunits that remain non-covalently associated at the surface of the viral particle (Hoffmann et al, 2020). Western blot analysis showed S2 was incorporated at comparable levels in cGAMP-S-VLPs and Empty-S-VLPs (Fig 7A). We collected serum samples 3 weeks after immunisation with cGAMP-S-VLPs and Empty-S-VLPs and analysed S-specific antibody titres using two different ELISA setups. The presence of cGAMP in VLPs enhanced the titres of antibodies recognising full-length S (Fig 7B) as well as its receptor-binding domain (RBD; Fig 7C), which interacts with the cellular receptor ACE2. We also determined the virus neutralisation

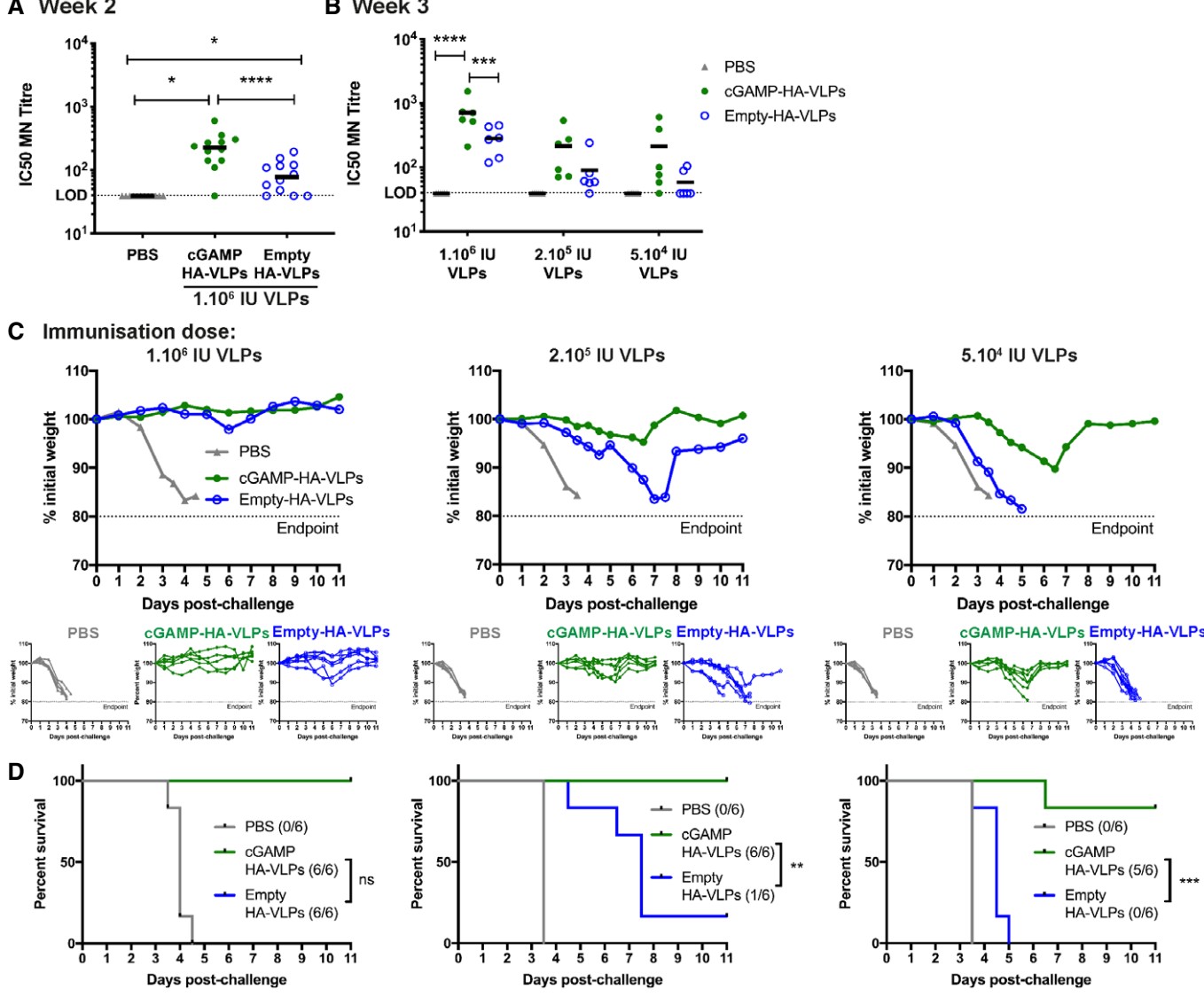

**Figure 5. cGAMP-VLPs pseudotyped with IAV HA induce neutralising antibodies and confer protection following IAV infection.**

C57BL/6 mice were immunised with PBS as a control, cGAMP-HA-VLPs or Empty-HA-VLPs *via* the intra-muscular route.

A, B  VLPs pseudotyped with IAV HA induce neutralising antibodies. Two (A) or three (B) weeks after immunisation with the indicated doses of VLPs, sera were collected, heat-inactivated, and titres of antibodies capable of neutralising an IAV expressing a matched HA protein were determined by microneutralisation (MN) assay. The dotted line shows the limit of detection (LOD).

C, D  Low doses of cGAMP-HA-VLPs confer protection following IAV challenge. One month after immunisation with the indicated doses of VLPs, animals were infected with $10^4$ TCID$_{50}$ of IAV PR8 virus. Weight loss was monitored over the following 11 days and is shown as a percentage of starting weight (C, upper graph shows mean and lower graphs show individual mice for each condition). Animals approaching the humane end-point of 20% weight loss were culled and survival to end-point curves are shown in (D).

Data information: In (A), data were pooled from two independent experiments including a total of 12 mice per condition. In (B–D), 6 mice per group were analysed for each VLP dose. In (A) and (B), symbols show data from individual animals. Horizontal lines indicate the mean. Statistical analyses were done using a Kruskal–Wallis test followed by Dunn's multiple comparisons test (A), a 2-way ANOVA followed by Tukey's multiple comparisons test (B) or a survival analysis with the log-rank (Mantel–Cox) test (D). ns $P \geq 0.05$; *$P < 0.05$; **$P < 0.01$; ***$P < 0.001$; ****$P < 0.0001$. See also Fig EV4.

---

capacity of antibodies induced after immunisation with VLPs containing S. We incubated live SARS-CoV-2 with serum samples and subsequently infected Vero cells. After overnight incubation, cells were fixed and stained with a nucleocapsid-specific antibody to reveal foci of infected cells (Fig EV5A). As a positive control, we used the RBD-specific and neutralising EY6A antibody (Zhou *et al*,

2020). As expected, EY6A reduced the number of infected foci (Fig EV5A and B). Similarly, sera from seven of twelve mice immunised with cGAMP-S-VLPs neutralised SARS-CoV-2 (Figs 7D and EV5A and B). In contrast, the serum from only four of twelve animals administered with Empty-S-VLPs reduced the infectivity of the virus, and samples from PBS-treated control mice had no effect. In

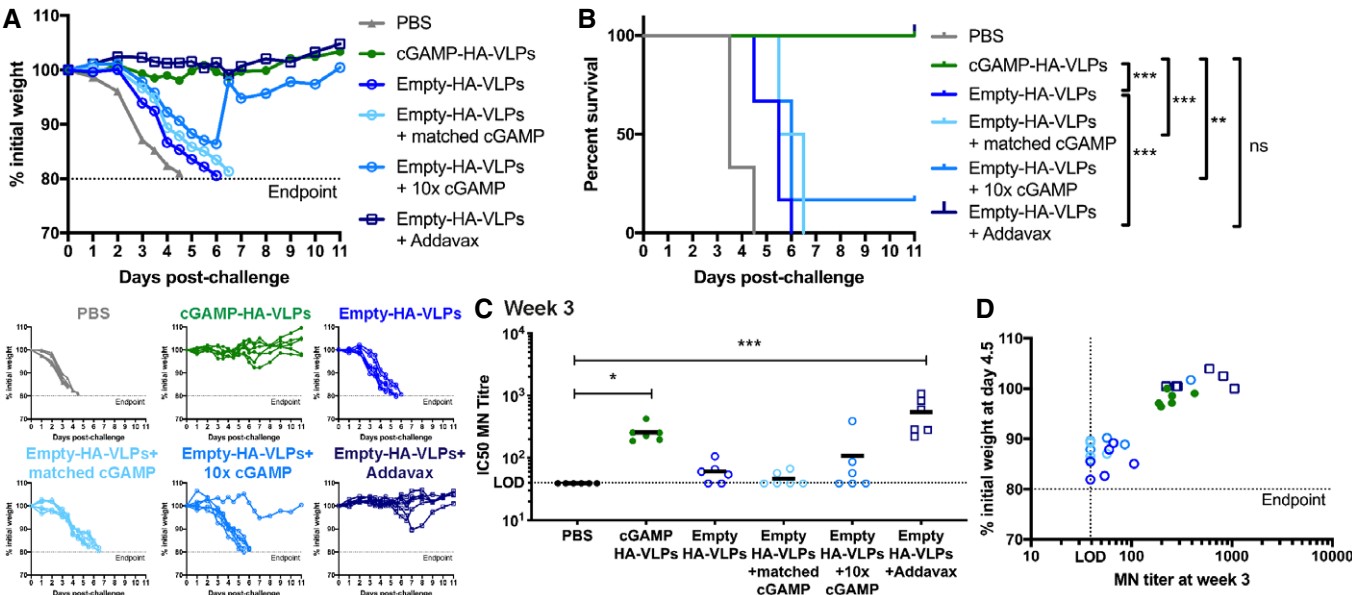

**Figure 6. Incorporation of cGAMP within VLPs is essential for its protective effect.**

Mice were immunised *via* the intra-muscular route with 5 × 10⁴ IU of cGAMP-HA-VLPs, Empty-HA-VLPs, Empty-HA-VLPs + matched cGAMP (3.5ng), Empty-HA-VLPs + 10× cGAMP (35 ng), Empty-HA-VLPs + AddaVax (1:1 vol ratio) or PBS as a control.

A, B   cGAMP-HA-VLPs and Empty-HA-VLPs + AddaVax protect against IAV challenge. One month after immunisation, animals were infected with 10⁴ TCID₅₀ of IAV PR8 virus. Weight loss was monitored over the following 11 days and is shown as a percentage of starting weight (A, upper graph shows mean and lower graphs show individual mice for each condition). Animals approaching the humane end-point of 20% weight loss were culled and survival to end-point curves are shown in (B).

C, D   cGAMP-HA-VLPs and Empty-HA-VLPs + AddaVax induce neutralising antibody responses that accompany protection. Three weeks after immunisation with 5 × 10⁴ IU of the indicated HA-VLPs, sera were collected and heat-inactivated. Titres of antibodies capable of neutralising an IAV expressing a matched HA protein were determined by microneutralisation (MN) assay (C). The dotted line shows the limit of detection (LOD). Correlation between the MN titres at week 3 and the weight loss at day 4.5 post-challenge is shown in (D).

Data information: 6 mice per group were analysed for each HA-VLP and PBS. In (C) and (D), symbols show data from individual animals. In (C), horizontal lines indicate the mean. Statistical analyses were done using a survival analysis with the log-rank (Mantel–Cox) test (B) or a Kruskal–Wallis test followed by Dunn's multiple comparisons test (C). ns $P \geq 0.05$; *$P < 0.05$; **$P < 0.01$; ***$P < 0.001$.

sum, these observations suggest that cGAMP loading of VLPs enhances the titres of SARS-CoV-2 neutralising antibodies upon immunisation.

# Discussion

New and/or more targeted adjuvants are needed for improved efficacy and safety of vaccines. There is growing interest in using adjuvants that specifically activate innate immune pathways used by cells to detect viral infections. cGAMP is one such example. cGAMP is a natural molecule produced by cells upon virus infection that specifically triggers STING, thereby inducing innate and adaptive immune responses. The vaccination strategy we describe here is based on coupling the adjuvant cGAMP with antigen(s) in a single entity, namely HIV-derived VLPs. We demonstrate that cGAMP loading of these VLPs increased CD4 and CD8 T-cell responses, as well as antibody responses, against protein antigens in the VLPs. As expected, the enhanced antibody response in mice immunised with cGAMP-VLPs was dependent on the expression of STING. Furthermore, vaccination with VLPs containing cGAMP protected mice against disease development following infection with a virus expressing a cognate antigen.

2′3′-cGAMP has attracted a lot of interest as an adjuvant since its identification in 2013 (Ablasser *et al*, 2013; Diner *et al*, 2013; Gao *et al*, 2013; Wu *et al*, 2013; Zhang *et al*, 2013). It is the CDN that is best recognised by all human STING variants, making it an ideal candidate adjuvant (Yi *et al*, 2013; Corrales *et al*, 2015). 2′3′-cGAMP has been tested pre-clinically in prophylactic vaccination models against infectious diseases and tumours, and as a therapeutic compound against tumours (Li *et al*, 2013; Blaauboer *et al*, 2015; Demaria *et al*, 2015; Temizoz *et al*, 2015; Lee *et al*, 2016; Li *et al*, 2016; Liu *et al*, 2016; Wang *et al*, 2016; Borriello *et al*, 2017; Takaki *et al*, 2017; Wang *et al*, 2017; Gutjahr *et al*, 2019; Luo *et al*, 2019; Vassilieva *et al*, 2019; Wang *et al*, 2020). These studies employed different vaccine formulations and delivery routes, ranging from co-injection with protein antigens or with inactivated virus to incorporation into nanoparticles that mimic pulmonary surfactant. In these reports, doses typically ranged from 1 to 20 μg per mouse. In contrast, we show here that incorporating cGAMP into VLPs was efficacious at doses as low as 3.5ng per mouse. An equivalent dose —or even a ten times excess—of soluble cGAMP co-injected with Empty-VLPs neither induced a neutralising antibody response to IAV nor protected against IAV challenge. cGAMP loading of VLPs therefore allows for a notable reduction in the dose of this adjuvant needed to induce protective immunity. Possible explanations for this

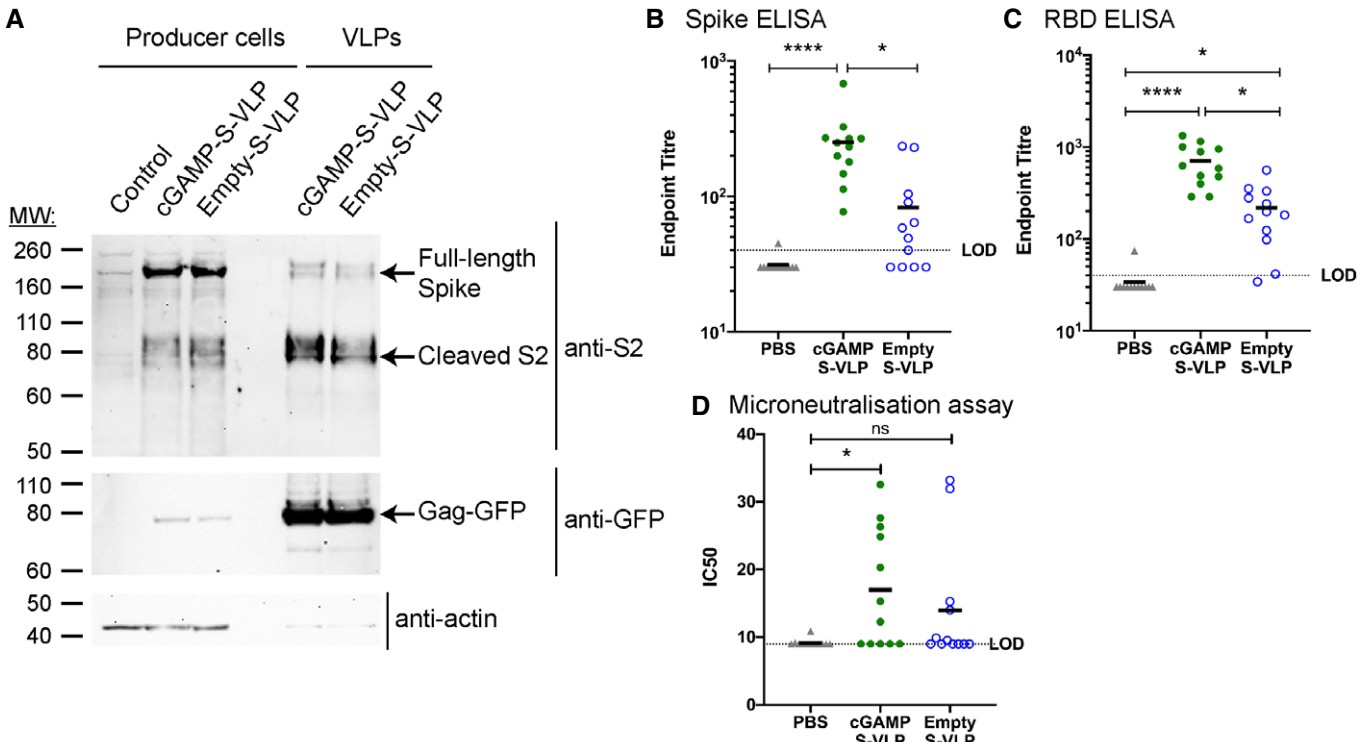

**Figure 7. cGAMP-VLPs containing the SARS-CoV-2 S protein induce an enhanced antibody response.**

A  cGAMP-S-VLPs and Empty-S-VLPs incorporate SARS-CoV-2 S. Lysates from VLP producer cells and VLP preparations were analysed by Western blot for the presence of the S2 subunit of SARS-CoV-2 S, Gag-GFP and actin using the indicated antibodies.

B, C  Immunisation with cGAMP-S-VLPs augments anti-Spike and anti-RBD antibody titres. Mice were immunised *via* the intra-muscular route with $5 \times 10^5$ IU of cGAMP-S-VLPs, Empty-S-VLPs or PBS as a control. Three weeks after immunisation, sera were collected, heat-inactivated, and titres of antibodies capable of binding to SARS-CoV-2 S (B) or its RBD (C) were determined by ELISA. The antibody response was expressed as end-point titre defined as the reciprocal of the highest serum dilution that gives a positive signal (blank+10SD). The dotted line shows the limit of detection (LOD).

D  Immunisation with cGAMP-S-VLPs induces neutralising antibodies. Using serum samples from (B), antibody titres capable of neutralising SARS-CoV-2 were determined by microneutralisation (MN) assay. Calculated IC50 doses from multiple serum dilutions are shown. The dotted line shows the LOD.

Data information: Panel (A) is representative of two independent experiments. In (B-D), data were pooled from two independent experiments including a total of 12 mice per condition; symbols show data from individual animals. Horizontal lines indicate the mean. Statistical analyses were done using a Kruskal–Wallis test followed by Dunn's multiple comparisons test (B, C, D). *$P < 0.05$; ****$P < 0.0001$. See also Fig EV5.

observation include that incorporating cGAMP inside viral particles (i) increases its stability at the site of injection by preventing degradation in the extracellular milieu (Li *et al*, 2014; Carozza *et al*, 2020) and/or (ii) prevents dilution by diffusion. Using small doses of cGAMP is likely to improve vaccine safety by limiting systemic inflammation. Indeed, we observed a transient weight loss of approximately 5% in mice receiving $10^6$ infectious units of cGAMP-VLPs (containing ~50 ng of cGAMP) or the adjuvants AddaVax and poly(I:C), but not in Empty-VLP-immunised mice or in animals receiving lower—but nonetheless efficacious—doses of cGAMP-VLPs. Future development of cGAMP-VLPs, building on our proof-of-concept study, will be required to assess and further improve safety and stability of cGAMP-loaded viral-vectored vaccines for testing in human. The HIV-derived VLPs we used are related to lentiviral vectors. This will allow application of established techniques and protocols for industrial production and purification and will likely accelerate this development phase (Merten *et al*, 2016).

In an effort to protect CDNs, some reports described systems where CDNs are incorporated into micro- or nanoparticles (Hanson

*et al*, 2015; Chen *et al*, 2018; Junkins *et al*, 2018; Wang *et al*, 2020). Although these studies showed a higher *in vivo* efficacy of CDNs upon incorporation into particles, they nonetheless employed high amounts of CDNs, typically above 1 µg. Furthermore, multiple immunisations were required, and only one study described a particle containing both antigen and CDN (Chen *et al*, 2018). In contrast, we demonstrate protective effects upon a single immunisation with VLPs containing both antigen and very low levels of cGAMP.

In addition to HIV-derived lentiviruses, other enveloped viruses also incorporate cGAMP (Bridgeman *et al*, 2015; Gentili *et al*, 2015). Therefore, our strategy of protecting the adjuvant cGAMP together with antigen in viral particles may be applicable to other viral-vectored vaccines such as modified vaccina virus Ankara (MVA). cGAMP loading of viral-vectored vaccines is likely to be advantageous not only in terms of safety but also by reducing cost of vaccine production, due to the requirement for lower doses. The latter is particularly important for lentivirus-based vectors that can typically only be produced at lower titres than other viral-vectored vaccines.

HIV-derived VLPs are a flexible system that allows incorporation of proteins of choice. We demonstrate this by decorating VLPs with IAV HA and SARS-CoV-2 S and show that upon immunisation, these VLPs induced antibodies that neutralised IAV expressing a matched HA protein and SARS-CoV-2, respectively. In future, other pathogen-derived proteins could be incorporated into cGAMP-loaded VLPs as a strategy to produce vaccines for a diverse breadth of pathogens. For example, multiple HA proteins from different IAV clades could be incorporated to induce broadly protective responses and envelope proteins from other recently emerging viruses such as Zika or Ebola could be delivered using this approach. Furthermore, we envisage that envelope glycoproteins from future emerging viruses with pandemic potential could be used in our system, allowing for rapid testing of cGAMP-VLPs containing these proteins as candidate vaccines.

It is noteworthy that Empty-VLPs were not inert but induced adaptive immune responses, albeit at lower levels than cGAMP-VLPs. This effect may be due to the presence in VLP preparations of nucleic acid fragments from producer cells, which could mediate a degree of adjuvanticity via receptors such as TLR9 (Pichlmair et al, 2007). Both splenic CD4 effector T-cell responses and Tfh cell numbers in draining lymphoid tissues were enhanced by incorporation of cGAMP in VLPs. It is likely that these effects explain the increased antibody responses we observed against VLP proteins. The CD4 T-cell response was skewed towards a Th1 phenotype, as indicated by robust IFNγ and TNFα production by CD4 T cells and enhanced IgG2a/c and IgG2b antibody responses. This is in line with previous studies showing that 2′3′-cGAMP induces a predominantly Th1-biased response with weak Th2 induction (Blaauboer et al, 2015; Borriello et al, 2017; Wang et al, 2020). Both Empty-VLPs and cGAMP-VLPs also induced IgG1 responses raising the possibility that cGAMP-VLPs trigger Th1 responses via cGAMP and Th2 responses via other VLP components. This may broaden the application of this platform and should be investigated in future studies.

Both the cell type mediating antigen presentation and the cytokines produced at the time of T-cell activation are crucial for polarisation of T-cell responses (O'Garra, 1998; Itano & Jenkins, 2003; Hong et al, 2018). Notably, IFN-I and IL-6 production by DCs have been reported to induce the development of Tfh cells in mice (Cucak et al, 2009; Nurieva et al, 2009; Riteau et al, 2016). We previously found that cGAMP-loaded viruses induce IFN-I in bone marrow-derived macrophages in vitro (Bridgeman et al, 2015). The activation of STING and down-stream IRF3 and NF-κB signalling by

cGAMP in vivo might therefore trigger production of IFN-I and IL-6 that could underlie the potent CD4 Tfh response elicited following immunisation with cGAMP-VLPs. The VLPs used here were pseudo-typed with VSV-G, which has a broad tropism (Finkelshtein et al, 2013; Hastie et al, 2013). Many cell types may therefore be infected at the site of injection (the muscle) and respond to cGAMP. Other types of VLPs such as Qβ-VLPs travel through the lymphatics to draining lymph nodes (Mohsen et al, 2017). Although larger than Qβ-VLPs, HIV-derived VLPs might also reach lymph nodes where they could infect other cell types. The specific cell types infected by VLPs in vivo and the cytokines induced by these cells are likely to be key aspects of the response induced by cGAMP-VLPs in vivo and warrant further investigation. It is also possible to replace VSV-G with other envelope proteins that target VLPs to specific cell types. For example, the envelope protein from Sindbis virus or antibodies such as those to DEC205 target virus particles to DCs, an essential antigen-presenting cell type (Trumpfheller et al, 2006; Yang et al, 2008). It will be interesting to determine whether DC-targeted VLPs containing cGAMP have a similar effect on the responses induced compared with the VSV-G pseudotyped VLPs described here. DC targeting could improve vaccine efficacy by restricting cGAMP delivery to relevant antigen-presenting cells.

Many studies are currently aimed at designing vaccines that induce antigen-specific CD8 T cells (Panagioti et al, 2018). We found that cGAMP loading of VLPs enhanced CD8 T-cell responses to the internal HIV-Gag antigen. However, the increased response in cGAMP-VLP-immunised mice did not result in a significant improvement in protection against a vaccinia virus expressing the same HIV-Gag compared with that observed in animals vaccinated with Empty-VLPs. The immunisation route and schedule employed here consisted of a single dose of VLPs injected intra-muscularly, which might not be adequate to elicit sufficiently high-magnitude CD8 T-cell responses to confer protection, but deployment in prime-boost regimens should be explored. In light of the neutralising antibody response induced by cGAMP-loaded VLPs, heterologous booster immunisations using non-particulate vaccines or viral particles with a different envelope protein should be considered in the future.

In summary, we provide proof-of-concept evidence that vaccination with HIV-derived VLPs containing both the adjuvant cGAMP and protein antigens constitutes an efficacious platform for induction of CD8 T-cell and neutralising antibody responses at low VLP doses. This VLP-based strategy of coupling adjuvant and antigen in a single entity is therefore a promising approach for future development of new and safer vaccines against a range of pathogens.

# Materials and Methods

**Reagents and Tools table**

| Reagent/Resource | Reference or source | Identifier or catalog number |
|---|---|---|
| **Experimental models** | | |
| HEK293T (*H. sapiens*) | Caetano Reis e Sousa lab | N/A |
| HEK293 (*H. sapiens*) | Caetano Reis e Sousa lab | N/A |
| 3C11 (*H. sapiens*) | Bridgeman et al (2015) | N/A |
| 143B (*H. sapiens*) | Nick Proudfoot lab | N/A |

**Reagents and Tools table**   (continued)

| Reagent/Resource | Reference or source | Identifier or catalog number |
|---|---|---|
| THP-1 (*H. sapiens*) | Vincenzo Cerundolo lab | N/A |
| MDCK-SIAT1 (*C. familiaris*) | ECACC Matrosovich et al (2003) | 05071502 |
| MDCK-PR8 (*C. familiaris*) | Powell et al (2012) | N/A |
| VERO (*C. aethiops*) | ATCC | CCL-81 |
| C57BL/6J (*M. musculus*) | Envigo | 057 |
| C57BL/6 (*M. musculus*) | University of Oxford Biomedical Services | N/A |
| C57BL/6 *Tmem173*$^{-/-}$ (*M. musculus*) | Jin et al (2011) | N/A |
| Influenza virus H1N1 A/Puerto Rico/8/1934 (Cambridge) (PR8) | Townsend lab. Virus produced in house via reverse genetics. Reference for the sequences: Winter et al (1981) | N/A |
| S-FLU vector expressing eGFP (S-eGFP) | Powell et al (2012) | N/A |
| Vaccinia virus vVK1 | Borrow et al (1994) | N/A |
| SARS-CoV-2 Victoria/01/2020 | Caly *et al* (2020) | N/A |
| **Recombinant DNA** | | |
| pGag-EGFP | NIH AIDS Reagent program | Cat #11468 |
| pCMV-VSV-G | Addgene | Cat #8454 |
| pcDNA3-Flag-mcGAS | Sun et al (2013) | N/A |
| pcDNA3-Flag-mcGAS-G198A/S199A | Sun et al (2013) | N/A |
| pcDNA3.1-H1 (PR8) | Original sequence from Winter et al (1981) | N/A |
| pcDNA3.1-Spike | Made in Townsend lab: codon-optimised Spike cDNA (genbank QHD43416.1) was synthesized by GeneArt and cloned into pcDNA3.1 | N/A |
| pNL4-3-deltaE-EGFP | NIH AIDS Reagent program | Cat #11100 |
| **Antibodies** | | |
| CD16/CD32 Rat anti-mouse<br>Clone: 93 | eBioscience | Cat #14-0161-82 |
| PE-Cy7 IFNγ rat anti-mouse<br>Clone: XMG1.2 | eBioscience | Cat # 25-7311-82 |
| BrilliantViolet 605 anti-mouse CD8a<br>Clone: 53-6.7 | Biolegend | Cat # 100743 |
| PerCP-Cy5.5 anti-mouse CD90.2<br>Clone: 30-H12 | Biolegend | Cat # 105337 |
| AlexaFluor 700 anti-mouse CD4<br>Clone: RM4-5 | Biolegend | Cat # 100536 |
| BrilliantViolet 510 anti-mouse MHC-II (I-A/I-E)<br>Clone: M5/114.15.2 | Biolegend | Cat # 107635 |
| PE anti-mouse TNFα<br>Clone: MP6-XT22 | Biolegend | Cat # 506305 |
| APC anti-mouse IL2<br>Clone: JES6-5H4 | Biolegend | Cat # 503809 |
| APC-Cy7 anti-mouse B220<br>Clone: RA3-6B2 | Biolegend | Cat # 103223 |
| BrilliantViolet 510 anti-mouse B220<br>Clone: RA3-6B2 | Biolegend | Cat # 103247 |
| PerCP-Cy5.5 anti-mouse IgD<br>Clone: 11-26c.2a | Biolegend | Cat # 405709 |
| AlexaFluor 647 GL7 | Biolegend | Cat # 144605 |
| PerCP-Cy5.5 anti-mouse CD44<br>Clone: IM7 | Biolegend | Cat # 103031 |
| BrilliantViolet 421 anti-mouse CXCR5<br>Clone: L138D7 | Biolegend | Cat # 145511 |
| APC anti-mouse PD1 | Biolegend | Cat # 109111 |

**Reagents and Tools table**   (continued)

| Reagent/Resource | Reference or source | Identifier or catalog number |
|---|---|---|
| Clone: RMP1-30 | | |
| PE anti-mouse CD95<br>Clone: Jo2 | BD Bioscience | Cat # 561985 |
| AlexaFluor 647 goat-anti mouse antibody | Life Technology | Cat # A21235 |
| HRP goat anti-mouse IgG1 | Bethyl laboratories | Cat # A90-205P |
| HRP goat anti-mouse IgG2a/c | Bethyl laboratories | Cat # A90-207P |
| HRP goat anti-mouse IgG2b | Bethyl laboratories | Cat # A90-109P |
| HRP goat anti-mouse IgM | Bethyl laboratories | Cat # A90-201P |
| HRP goat anti-mouse | Agilent | Cat # P0447 |
| Anti-IAV Hemagglutinin H1 & H5<br>Clone: 21-D8-5A | Xiao et al (2018) | N/A |
| Anti-SARS-CoV-2 Spike (S2 subunit)<br>Clone: FD-10A | Townsend lab, Huang et al (2021) | N/A |
| Mouse anti-GFP | Roche | Cat # 11814460001 |
| HRP anti-actin | Sigma | Cat # A3854 |
| **Chemicals, Enzymes and other reagents** | | |
| LIVE/DEAD fixable violet dead cell stain | ThermoFischer scientific | Cat # L34955 |
| LIVE/DEAD fixable aqua dead cell stain | ThermoFischer scientific | Cat # L34957 |
| HIV-1 Con B Gag Peptide Set | NIH AIDS Reagent Program | Cat # 8117 |
| pep 92 9-mer (HIV-SQV) | Genscript | Custom synthesis |
| pep 92 10-mer | Genscript | Custom synthesis |
| pep 92 11-mer | Genscript | Custom synthesis |
| Recombinant HIV-1 IIIB pr55 Gag protein | NIH AIDS Reagent Program | Cat # 3276 |
| Recombinant VSV-G protein | alpha diagnostic international | Cat # VSIG15-R-10 |
| 2'3'-cGAMP | Invivogen | Cat # tlrl-nacga23 |
| 2'3'-cGAMP VacciGrade | Invivogen | Cat # vac-nacga23 |
| Poly(I:C) (HMW) | Invivogen | Cat # tlrl-pic |
| AddaVax | Invivogen | Cat # vac-adx-10 |
| Brilliant stain buffer | BD Bioscience | Cat # 563794 |
| Fixation/Permeabilisation Solution kit with GolgiStop | BD Bioscience | Cat # 554715 |
| eBioscience FoxP3/transcription factor staining Buffer Set | ThermoFischer scientific | Cat # 00-5523-00 |
| BD Cellfix | BD Bioscience | Cat # 340181 |
| Fugene 6 | Promega | Cat # E2691 |
| Lipofectamine 2000 | ThermoFischer scientific | Cat # 11668030 |
| One-Glo luciferase assay system | Promega | Cat # E6120 |
| Amicon Ultra 3K filter columns | Millipore | Cat # UFC500396 |
| 2'-3' cGAMP ELISA kit | Cayman chemical | Cat # 501700 |
| Mouse IFNγ ELISPOT BASIC (ALP) kit | Mabtech | Cat # 3321-2A |
| Mouse IgG Basic ELISPOT BASIC (ALP) kit | Mabtech | Cat # 3825-2A |
| Red blood cell lysis buffer | Sigma | Cat # R7757-100ML |
| TMB substrate | Invitrogen | Cat # 00-4201-56 |
| BM Blue POD substrate | Roche | Cat # 11484281101 |
| TPCK-treated trypsin | Sigma | Cat # T1426 |
| **Software** | | |
| FlowJo v10.7.1 | | |
| Graphpad Prism v8.4.3 | | |

                                                                    

## Methods and Protocols

### Mice

All mice were on the C57BL/6 background. Only female mice between 6 and 8 weeks old were used. Animals were housed in individually ventilated cages and standard husbandry conditions were used. WT mice were obtained from University of Oxford Biomedical Services or Envigo RMS (UK) Limited. $Tmem173^{-/-}$ (STING-deficient; herein referred to as $Sting^{-/-}$ for simplicity) mice were a gift from J Cambier (Jin et al, 2011). This work was performed in accordance with the UK Animals (Scientific Procedures) Act 1986 and institutional guidelines for animal care. This work was approved by project licences granted by the UK Home Office (PPL No. 40/3583, No. PC041D0AB and No. PBA43A2E4) and was also approved by the Institutional Animal Ethics Committee Review Board at the University of Oxford.

### Cells

Cell lines (HEK293T, HEK293, 3C11, 143B, MDCK-SIAT1, MDCK-PR8, VERO) were maintained in DMEM (Sigma-Aldrich) supplemented with 10% FCS (Sigma-Aldrich) and 2 mM L-glutamine (Gibco) at 37°C and 5% $CO_2$. 3C11 cells are HEK293 cells stably transduced with an ISRE-Luc reporter construct (Bridgeman et al, 2015). 143B cells were a kind gift from N. Proudfoot (University of Oxford).

Bone marrow cells were isolated from humanely killed adult mice by standard protocols and grown in 6-well plates for 5 days in RPMI supplemented with 10% FCS, 2 mM L-glutamine, 1% PenStrep and 20 ng/ml mouse GM-CSF to obtain bone marrow-derived myeloid cells (BMMCs).

### Reagents and antibodies

See Reagents and Tools Table.

### VLP and HIV-1 vector production

All VLPs were produced by transient transfection of HEK293T cells with Fugene 6. HEK293T were seeded into 15-cm dishes to reach 60–70% confluency the next day, and VLPs were produced by co-transfecting plasmids encoding Gag-eGFP and the VSV-G envelope (pGag-EGFP and pCMV-VSV-G, respectively) at a ratio of 2:1. VLPs were loaded with cGAMP by co-transfecting at the same time a plasmid encoding mouse cGAS WT (pcDNA3-Flag-mcGAS). Empty-VLPs were produced as control by co-transfecting a catalytically inactive mouse cGAS (cGAS AA; pcDNA3-Flag-mcGAS-G198A/S199A). One day after transfection, the medium was changed. Supernatants were collected 24, 32 and 48 h after medium change, centrifuged and filtered (cellulose acetate membrane 0.45 μm pore-size). At each media change, VLPs were concentrated by ultracentrifugation through a 20% sucrose cushion at 90,000 g for 2.5 h at 8°C using a Beckman SW32 rotor. VLPs were resuspended in PBS, and subsequent harvests were resuspended using the resuspended VLPs from previous harvests to maximise titre.

For pseudotyping cGAMP-VLPs and Empty-VLPs with influenza haemagglutinin H1 (HA; pcDNA3.1-H1 (PR8)), cells were transfected as above with the following plasmids: Gag-eGFP, VSV-G, HA and cGAS WT or AA at a ratio of 2:1:1:2. For pseudotyping cGAMP-VLPs and Empty-VLPs with SARS-CoV-2 Spike (S; pcDNA3.1-Spike), cells were transfected with the following plasmids: Gag-eGFP, VSV-G, S and cGAS WT or AA at a ratio of 2:1:1:2.

To produce VSV-G pseudotyped HIV-1 vectors for neutralisation assays, HEK293T cells were co-transfected with the following plasmids: HIV-1 NL4-3 ΔEnv GFP (pNL4-3-deltaE-EGFP) and VSV-G at a ratio of 2:1.

### VLP titration and cGAMP incorporation assays

HEK293 cells were seeded at a density of $1 \times 10^5$ cells per well in 24-well plates. The next day, cells were infected with decreasing amounts of VLPs in the presence of 8 μg/ml of polybrene. 24 h after infection, cells were collected and first stained with anti-CD16/32 and Aqua fixable Live/Dead in FACS Buffer (PBS, 1% FCS, 2mM EDTA) for 15 min at RT. Cells used for titration of HA-VLPs were also stained for HA using a primary human anti-H1 & H5 antibody (clone 21-D8-5A) in FACS Buffer for 30 min at 4°C. Cells were then washed twice and further stained with a secondary goat anti-human Alexa Fluor 647-conjugated antibody for 30 min at 4°C, followed by two washes. All cells were fixed using BD Cellfix before acquisition on an Attune Nxt flow cytometer. Infection was measured by analysing GFP-positive cells by flow cytometry using FlowJo version 10. VLP titres were calculated based on the number of GFP$^+$ cells compared with the number of cells in the well at the time of infection and expressed as infectious units/ml (IU/ml). Supernatants from infected cells were transferred onto ISRE reporter cells to assess IFN-I production in response to cGAMP incorporated in VLPs as described previously (Bridgeman et al, 2015). After 24 h of incubation with supernatants, expression of the ISRE-Luc reporter was assessed using the One-Glo luciferase assay system.

Small molecular extracts were prepared from VLPs as described (Mayer et al, 2017). Briefly, $2 \times 10^6$ IU of ultra-centrifuged VLPs resuspended in PBS were lysed in X-100 Buffer (1 mM NaCl, 3 mM $MgCl_2$, 1mM EDTA, 1% Triton X-100, 10 mM Tris pH7.4) by adding 1/10 volume of 10× buffer for 20 min on ice while vortexing regularly. After centrifugation at 1,000 g for 10 min at 4°C, supernatants were treated with 50 U/ml of benzonase for 45 min on ice. Samples were then extracted with phenol–chloroform, and the aqueous phase was then transferred to Amicon Ultra 3K filter columns. After filtering by centrifugation at 14,000 g for 30 min at 4°C, samples were dried in a SpeedVac and resuspended in 200 μl of water. cGAMP was quantified using the 2′–3′ cGAMP ELISA kit following the manufacturer's instructions and the cGAMP bioassay as described previously (Bridgeman et al, 2015). Briefly, THP-1 cells were seeded in the presence of 5 ng/ml PMA. The next day, small molecular extracts were delivered to the cells using a permeabilisation buffer (2× PERM: 100 mM HEPES-HCl (pH 7.4), 200 mM KCl, 6 mM $MgCl_2$, 0.4% BSA, 170 mM sucrose, 2 mM ATP, 0.2 mM GTP, 0.002% digitonin) alongside a cGAMP standard. After 30 min of incubation at 37°C, cells were washed, fresh medium was added and the cells were placed back in the incubator for an additional 24 h. IFN-I produced was quantified using an ISRE reporter assay and the quantity of cGAMP was calculated using the cGAMP standard. For both assays, a small molecular extract of a control sample containing 1 μg of chemically synthesised cGAMP was used to normalise the quantities of cGAMP in the VLP samples.

### IAV

The influenza virus H1N1 A/Puerto Rico/8/1934 (Cambridge) (PR8) and the non-replicating S-FLU vector expressing eGFP (S-eGFP) were generated as previously described (Powell et al, 2012).

Plasmids encoding the IAV Cambridge strain of A/Puerto Rico/8/34 were used to generate the wild-type H1N1 A/Puerto Rico/8/1934 (Cambridge) (PR8) seed virus. The same plasmids were used to generate the PR8 S-eGFP with slight modifications: the HA coding region in the plasmid expressing HA viral RNA was replaced with eGFP and an additional plasmid was included to provide a functional PR8 HA in *trans* to rescue the PR8 S-eGFP seed virus. Briefly, the plasmids were transfected into HEK293T cells using Lipofectamine 2000 and supernatant containing seed virus was collected 72 h after transfection. The wild-type PR8 virus and the S-eGFP vector were then propagated by infecting MDCK-SIAT1 cells or MDCK-SIAT1 stably transfected with PR8 HA (MDCK-PR8), respectively, with seed virus, followed by medium change into VGM (Viral Growth Media; DMEM, 1% BSA, 10 mM HEPES buffer, 1% PenStrep) containing 1 µg/ml TPCK-treated trypsin. Viruses were harvested 48 h later. The $TCID_{50}$ was determined by infecting MDCK-SIAT1 or MDCK-PR8 cells with a ½-log dilution series of viruses in VGM for 1 h in eight replicates using 96-well flat-bottom plates. Next, 150µl per well of VGM with TPCK-treated trypsin (1 µg/ml) was added and cells were further incubated for 48 h at 37°C. The PR8 virus and the S-eGFP vector were quantified by nucleoprotein (NP) staining and eGFP expression, respectively, and $TCID_{50}$ was calculated using the method of Reed and Muench (Reed & Muench, 1938).

### Vaccinia virus

Stocks of the vaccinia virus expressing HIV-1 HXB.2 Gag (vVK1) were produced by growth in 143TK cells, and infectious virus titres were determined by plaque assay (Borrow *et al*, 1994).

### Immunisation and viral challenge of mice

Animals were injected intra-muscularly with 50 µl per hindleg of PBS or $10^6$ IU of cGAMP-VLPs or Empty-VLPs, unless otherwise stated, under inhalation isoflurane (IsoFlo, Abbott) anaesthesia. For comparisons with other adjuvants, when indicated, mice were immunised with $10^6$ IU of Empty-VLPs mixed with 25 µg poly(I:C) (HMW), or with $5 \times 10^4$ IU of Empty-VLPs with AddaVax (1:1 v/v). Weight was monitored every day for 14 days. For immunophenotyping, mice were culled on day 14 by inhalation of carbon dioxide and cervical dislocation. For viral challenge experiments, mice were monitored every other day for an additional 2 weeks before challenge.

For IAV challenge, blood samples were acquired 2 weeks after immunisation for evaluation of the serum antibody response. Mice were then challenged a month after immunisation *via* the intranasal route with 10,000 $TCID_{50}$ of PR8 diluted in 50 µl VGM under inhalation isoflurane anaesthesia. Weight was monitored daily, and mice were culled by inhalation of carbon dioxide and cervical dislocation when body weight loss approached the humane end-point of 20%.

For vaccinia virus challenge, mice were infected *via* the intraperitoneal route with $10^6$ PFU vVK1 in 100 µl PBS. Weight was monitored daily for 5 days. Animals were then culled by inhalation of carbon dioxide and cervical dislocation, and ovaries were collected for virus titration.

### Analysis of T-cell responses by ICS and ELISPOT

Splenocytes were obtained by separating spleens through a 70-µm strainer and were then treated with red blood cell lysis buffer for 5 min, washed and resuspended in RPMI supplemented with 2% human serum, 2 mM L-glutamine, 1% PenStrep (R2).

For ELISPOT assays, splenocytes were seeded in R2 at a density of $1.5 \times 10^5$ cells per well on ELISPOT plates pre-coated with anti-IFNγ detection antibody. Cells were either non-treated or treated with 2 µg/ml HIV-1 Gag peptide or with 10 ng/ml PMA and 1 µg/ml ionomycin as a control, and incubated for 48 h at 37°C before detection according to the manufacturer's instructions (Mouse IFNγ ELISPOT BASIC (ALP) kit).

For intracellular cytokine staining (ICS), cells were seeded in R2 at a density of $1 \times 10^6$ cells per well in a round-bottom 96-well plates. Cells were either non-treated or treated with 2 µg/ml HIV-SQV 9-mer peptide or co-cultured with BMMCs pulsed overnight with cGAMP-VLP at a multiplicity of infection of 1. Cells were also treated with 10 ng/ml PMA and 1 µg/ml ionomycin as a positive control. After 1 h of incubation at 37°C, Golgi STOP was added according to manufacturer's instructions. After a further 5 h of incubation at 37°C, cells were washed twice in FACS buffer (PBS, 1% FCS, 2mM EDTA), incubated with anti-CD16/32 and Aqua or violet fixable Live/Dead in FACS Buffer for 15 min at RT and were then washed twice in FACS Buffer. Subsequent extracellular staining involved incubation of cells for 30 min at 4°C with the following antibodies: anti-CD8 BV605 and anti-CD90.2 PerCP-Cy5.5 in FACS Buffer for CD8 T-cell analysis in cells stimulated with the HIV peptide, or anti-CD4 AF700, anti-CD8 BV605 and anti-MHC-II BV510 in Brilliant stain buffer for CD4 T-cell analysis in cells stimulated with pulsed BMMCs. Cells were then washed twice in FACS Buffer and fixed using BD Cytofix/Cytoperm buffer for 20 min at 4°C. After two washes in FACS Buffer with 10% BD Cytoperm/wash, intracellular staining was performed for using anti-TNFα PE, anti-IFNγ PE-Cy7 and anti-IL2 APC in FACS Buffer with 10% BD Cytoperm/wash for 30 min at 4°C. After two washes in FACS Buffer with 10% BD Cytoperm/wash, cells were fixed for 10 min at RT in BD Cellfix, washed again and resuspended in FACS Buffer for acquisition on Attune NxT flow cytometers. Analysis was performed using FlowJo version 10. Gates for phenotypic markers of CD4 and CD8 T cells were based on FMO controls. Unstimulated control cells were used for other gates.

### Analysis of serum antibody titres by ELISA
#### Anti-VLP, anti-VSV-G and anti-Gag ELISAs

To extract protein, VLPs were lysed in PBS containing 0.5% Triton X-100 and 0.02% sodium azide for 10 min at RT. Quantity of protein extracted was quantified by BCA assay.

Costar high-binding half-area flat-bottom 96-well plates were coated overnight at 4°C with either 10 µg/ml unconcentrated cGAMP-VLP lysates, 1 µg/ml concentrated cGAMP-VLP lysates, 0.5 µg/ml recombinant HIV-1 IIIB pr55 Gag protein or 1.5 µg/ml recombinant VSV-G protein. The next day, plates were washed twice in PBS, then twice in PBS with 0.1% Tween-20 (wash buffer) and blocked in PBS with 3% BSA for 2 h at RT. Sera collected on day 14 after immunisation were serially diluted in PBS with 0.5% BSA starting at a dilution of 1/200 and diluting 1/3. After four washes in wash buffer, serum dilutions were added to the plates in duplicates (25 µl per well) and incubated for 1 h at 37°C. Plates were washed four times in wash buffer. Next, 50 µl per well of HRP-conjugated antibodies recognising different antibody classes or

subclasses was added using the following dilutions: goat anti-mouse IgG1 / IgG2a/c / IgG2b, 1/10,000; IgM, 1/2,000 in PBS with 0.5% BSA and incubated for 1 h at RT. Plates were washed four times in wash buffer, and 50 μl of TMB substrate was added per well. Plates were incubated for approximately 30 min or until the signal was saturating, and 50 μl of STOP solution was added per well before reading absorbance at 450 and 570 nm on a CLARIOstar plate reader.

### SARS-CoV-2 S ELISA

To detect antibodies binding to SARS-CoV-2 S, we created MDCK-S cells by stably transducing parental MDCK-SIAT1 cells with a lentiviral vector expressing full-length SARS-CoV-2 S (Huang *et al*, 2021). MDCK-S cells were seeded into flat-bottom 96-well plates ($3 \times 10^4$ cells per well) and incubated at 37°C overnight. Cells were then washed twice with PBS. Heat-inactivated mouse sera were serially diluted in PBS/0.1% BSA, and 50 μl was added to the cells for 1 h at RT. Plates were then washed twice with PBS, and 50 μl AlexaFluor 647-conjugated goat anti-mouse antibody (1:500 in PBS/0.1% BSA) was added per well for 1 h at RT. Plates were then washed twice with PBS, and 100 μl of PBS/1% formalin was added per well. Fluorescence signals were acquired on a CLARIOstar plate reader. The S-specific antibody response was calculated as end-point titre (EPT). EPT is defined as the reciprocal of the highest serum dilution that gives a positive signal (blank+10SD) determined using a five-parameter logistic equation.

### SARS-CoV-2 RBD ELISA

NUNC plates were coated with 50 μl purified RBD-His$_6$ (2 μg/ml in PBS) at 4°C overnight. Plates were then washed with PBS and blocked with 300 μl of 5% skimmed milk for 1 h at RT. Blocked plates were then washed with PBS. Heat-inactivated mouse sera were serially diluted in PBS/0.1% BSA, and 50 μl was transferred to the plates for 1 h at RT. Plates were then washed with PBS, and 50 μl of secondary HRP-conjugated goat anti-mouse antibody (1:800 in PBS/0.1% BSA) was added per well for 1 h at RT. Plates were washed with PBS, and 50 μl of BM Blue POD substrate was added for 5 min. The reaction was stopped by adding 50 μl of 1 M $H_2SO_4$. OD450 was read on a CLARIOstar plate reader. EPT was determined as above. This ELISA to detect RBD-specific antibodies is described in detail in Huang *et al* (2021).

### Analysis of germinal centre B cells and T follicular cells in draining lymph nodes

Both inguinal lymph nodes were meshed through a 70-μm strainer. $10^6$ cells per animal were used for each staining.

For germinal centre B-cell analysis, cells were first stained with anti-CD16/32 and Aqua fixable Live/Dead in FACS Buffer for 15 min at RT. After two washes in FACS Buffer, extracellular staining was performed using anti-B220 APC-Cy7, anti-CD95 PE, anti-IgD PerCP-Cy5.5 and GL7 AF647 in FACS Buffer for 30 min at 4°C. Cells were then washed, fixed for 10 min at RT in BD Cellfix, washed again and resuspended in FACS Buffer.

For T follicular cell analysis, cells were first stained with anti-CD16/32 and Aqua fixable Live/Dead in FACS Buffer for 15 min at RT. After two washes in FACS Buffer, extracellular staining was performed using anti-B220 BV510, anti-CD4 AF700, anti-CD44

PerCP-Cy5.5, anti-CXCR5 BV421 and anti-PD-1 APC in Brilliant stain buffer for 1 h at 4°C. After two washes in FACS Buffer, cells were fixed using the eBioscience FoxP3 fixation buffer for 25 min at RT. Cells were then washed twice in cold eBioscience Perm buffer, and intracellular staining was performed using anti-FoxP3 PE-Cy7 in eBioscience Perm buffer for 40 min at RT. After two washes in eBioscience Perm buffer, cells were then fixed for 10 min at RT in BD Cellfix, washed again and resuspended in FACS Buffer for acquisition on Attune NxT flow cytometers. Analysis was performed using FlowJo version 10. Gates for phenotypic markers of CD4 T cells and B cells were based on FMO controls, and gates for GC and Tfh/Tfr markers were based on PBS-immunised mice.

### B cell ELISPOT

Cells from spleen and draining lymph nodes were collected as described above and counted. Three different amounts of cells ($10^6$, $3 \times 10^5$, $1 \times 10^5$) were seeded in duplicate in R2 on ELISPOT plates coated overnight with 2 μg/ml of lysates from cGAMP-VLPs (see ELISA). Plates were then incubated overnight at 37°C before detection according to manufacturer's instruction (Mouse IgG Basic ELISPOT BASIC (ALP) kit). Analysis was performed using the cell density showing the least background in PBS-injected mice.

### IAV microneutralisation assay

Microneutralisation (MN) assay was performed as described (Powell *et al*, 2012) with minor modifications. Briefly, a single cycle IAV expressing eGFP (S-eGFP (PR8)) containing the H1 haemagglutinin was titrated to give saturating infection of $3 \times 10^4$ MDCK-SIAT1 cells per well in 96-well flat-bottom plates, detected by eGFP fluorescence. Murine sera were heat inactivated for 30 min at 56°C. Dilutions of sera were incubated with S-eGFP for 2 h at 37°C before addition to $3 \times 10^4$ MDCK-SIAT1 cells per well. Cells were then incubated overnight before fixing in 4% formaldehyde. The suppression of infection was measured on fixed cells by fluorescence on a CLARIOstar fluorescence plate reader.

### SARS-CoV-2 MN assay

Heat-inactivated mouse sera were serially diluted in DMEM+1% FBS and then incubated with live SARS-CoV-2 for 90 min at RT, using a concentration of virus that yields approximately 100 foci per well. $4.5 \times 10^4$ Vero ATCC (CCL81) cells were then added to each well. After 2 h of incubation, the cells were overlaid with DMEM+1% FBS containing CMC at a final concentration of 1.5%. Cells were then incubated for 20 h before fixation with 4% PFA for 30 min at RT. Cells were then permeabilised with PBS+2% Triton X-100 for 30 min at 37°C before staining with an antibody targeting the SARS-CoV-2 Nucleocapsid (clone EY-2A) for 1 h at RT, followed by an HRP-coupled anti-human IgG antibody. Infected cells were revealed using TrueBlue peroxidase substrate incubated for 10 min at RT. Plates were then washed with water and counted with an ELISPOT plate reader. The number of spots per well was normalised to wells that received no serum and the IC50 was calculated in Prism using a non-linear regression [inhibitor] vs normalised response—variable slope. This work was performed in a BSL3 laboratory and details of the MN assay will be described in Harding, A. & Gilbert-Jaramillo, J. *et al* (manuscript in preparation).

### Western blot

Cells were lysed in NP-40 buffer (150 mM NaCl, 1% NP-40, 50 mM Tris pH 8.0) with protease inhibitors. After 20 min of incubation on ice, lysates were centrifuged at 17,000 $g$ for 10 min at 4°C. Supernatant was collected and diluted with sample buffer containing beta-mercaptoethanol before denaturation at 95°C for 5 min. Samples were loaded on pre-cast 4–12% gradient Bis-Tris protein gels that were run with MOPS buffer at 120 volts for 2 h. Transfer to nitrocellulose membranes was performed in transfer buffer (25mM Tris, 192 mM glycine, 10% methanol) at 90 volts for 2 h. Membranes were blocked in 5% milk powder in Tris-buffered saline with 0.05% NP-40 (TBSN) for 1 h at room temperature, then washed five times for 5 min in TBSN. Membranes were incubated with primary antibody in 5% milk TBSN overnight at 4°C and washed five times for 5 min in TBSN. Membranes were then incubated with HRP-coupled secondary antibody in 5% milk TBSN for 1 h at room temperature and washed five times for 5 min in TBSN. Western Lightning Plus-ECL substrate was used for signal detection on an iBright FL1000 machine. In some experiments, membranes were stripped with 0.2 M glycine, 1% SDS at pH 2.5 for 15 min, washed, blocked and re-probed with a different antibody.

### Vaccinia virus plaque assay

Ovaries collected in D0 (DMEM, 1% PenStrep) were homogenised using glass beads in screw cap tubes in a homogeniser (two cycles at speed 6.5 for 30 s). Samples were then placed on ice for 1–2 min, and homogenisation was repeated. Samples were then subjected to three freeze-thaw cycles between 37°C and dry ice and sonicated three times for 30 s with 30-s intervals on ice. Supernatants containing virus were collected in new tubes after centrifugation at 9,600 $g$ for 3 min at 4°C.

143B cells were seeded in 12-well plates at a density of $0.25 \times 10^6$ cells per well in 1 ml D10. The next day, log serial dilutions of virus-containing samples were prepared in D0. Supernatant was replaced with 550 μl of diluted virus-containing samples and incubated for 2 h at 37°C, swirling plates every 30 min to avoid drying. Virus containing samples were then removed, and cells were covered in 1.5 ml of D10 containing 1% Pen/Strep and 0.5% carboxymethylcellulose (CMC). Forty-eight hours after infection, cells were carefully washed with PBS and fixed in 4% formaldehyde for 20 min at RT before staining with 0.5% crystal violet.

### Statistics

Statistical analysis was performed in GraphPad Prism v7.00 as detailed in the figure legends.

## Data availability

The authors declare that all data supporting the findings of this study are available within the paper and its associated files. No primary data sets have been generated or deposited.

Expanded View for this article is available online.

## Acknowledgements

The authors thank William James for providing access to the BSL3 facility and for his support in developing the MN assay for SARS-CoV-2. We further thank Andrew McMichael, Adrian Hill, Michelle Linterman, Daniel Radtke, Oliver Bannard, Nicolas Manel, Rachel E. Rigby and members of the Rehwinkel lab for discussion. The authors thank Uzi Gileadi and Vincenzo Cerundolo for their help with IAV infections. The authors thank Nicholas Proudfoot and Bernard Moss for providing, respectively, the 143B cells and the vVK1 vaccinia virus. The following reagents were obtained through the NIH AIDS Reagent Program, Division of AIDS, NIAID, NIH: HIV-1 Con B Gag Peptide Set, HIV-1 HXB2 Gag-EGFP Expression Vector (Cat#11468) from Dr. Marilyn Resh, HIV-1 NL4-3 ΔEnv EGFP Reporter Vector from Drs. Haili Zhang, Yan Zhou, and Robert Siliciano (cat# 11100), HIV-1IIIB pr55 Gag. This work was funded by the UK Medical Research Council [MRC core funding of the MRC Human Immunology Unit, MC_UU_00008/1; J.R., J.F., H.D. and MRC Programme grant, MR/K012037; P.B.], the Wellcome Trust [grant number 100954; J.R.], and the NIH, NIAID, DAIDS [UM1 grants AI00645 (Duke CHAVI-ID) and AI144371 (Duke CHAVD); P.B.]. P.B. is a Jenner Institute Investigator. J.G-J. is funded by the National Ecuadorian Governemnt—Secretaría Nacional de Educación Superior, Ciencia, Tecnologia y Educación—SENESCYT. M.L.K. is funded by the Biotechnology and Biological Sciences Research Council (BBSRC) [grant number BB/M011224/1]. R.A.R. acknowledges the generous support of philanthropic donors that allowed funding from the University of Oxford's COVID-19 Research Response Fund, which also supported the SARS-CoV-2 BSL3 facility and microneutralisation assay. Initial funding for the Virus Screening Facility was provided by the Oxford BRC and Cancer Research UK. The funders had no role in study design, data collection and analysis, decision to publish or preparation of the manuscript.

## Author contributions

LC, AB and JR conceptualised the study; LC, AB, TKT, PR, JF, IP-P, TP, JG-J, MLK, XL and RAR contributed to methodology; n.a. provided software; LC and JR validated the study; LC and JR involved in formal analysis; LC, AB, TKT and JF investigated the study; RB and PB provided resources; LC curated the data; LC and JR wrote—original draft; all authors wrote—review & editing; LC and JR visualised the study; JR, AT, HD and PB supervised the study; LC involved in project administration; JR contributed to funding acquisition.

## Conflict of interest

The authors declare that they have no conflict of interest.

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
