## [Review Process File · EMBO Reports]

Inclusion of cGAMP within virus-like particle vaccines enhances their immunogenicity

Lise Chauveau, Anne Bridgeman, Tiong Kit Tan, Ryan Beveridge, Joe Frost, Pramila Rijal, Isabela Pedroza-Pacheco, Thomas Partridge, Javier Gilbert-Jaramillo, Michael Knight, Xu Liu, Rebecca Russell, Persephone Borrow, Hal Drakesmith, Alain Townsend, and Jan Rehwinkel
DOI: [10.15252/embr.202152447](https://doi.org/10.15252/embr.202152447)

Corresponding author(s): Jan Rehwinkel (jan.rehwinkel@imm.ox.ac.uk)

Review Timeline:	Submission Date:	12th Jan 21
	Editorial Decision:	13th Jan 21
	Revision Received:	1st Apr 21
	Editorial Decision:	3rd May 21
	Revision Received:	21st May 21
	Accepted:	26th May 21

Editor: Achim Breiling

Transaction Report: This manuscript was transferred to EMBO reports following peer review at EMBO Molecular Medicine.

Dear Prof. Rehwinkel,

Thank you for transferring your manuscript to EMBO reports. I now went through your manuscript and the referee reports from EMBO Molecular Medicine (attached below). The referees acknowledge that the findings are of interest. Nevertheless, they have raised a number of concerns and suggestions to improve the manuscript, or to strengthen the data and the conclusions drawn.

EMBO reports emphasizes novel functional over detailed mechanistic insight, but asks for strong in vivo relevance of the findings, and clear experimental support of the major conclusions. Thus, we will not require addressing points regarding more mechanistic details experimentally. However, it will be necessary that in a revised manuscript you address all points questioning the main conclusions of the study, and all technical concerns, or points regarding the experimental design, model systems used, or data presentation.

Given the constructive referee comments, we would like to invite you to revise your manuscript with the understanding that all referee concerns must be addressed in the revised manuscript and/or in a detailed point-by-point response. Acceptance of your manuscript will depend on a positive outcome of a second round of review. It is EMBO reports policy to allow a single round of revision only and acceptance of the manuscript will therefore depend on the completeness of your responses included in the next, final version of the manuscript.

Revised manuscripts should be submitted within three months of a request for revision. We are aware that many laboratories cannot function at full efficiency during the current COVID-19/SARS-CoV-2 pandemic and we have therefore extended our 'scooping protection policy' to cover the period required for full revision. Please contact me to discuss the revision should you need additional time, and also if you see a paper with related content published elsewhere.

- 1) a .docx formatted version of the final manuscript text (including legends for main figures, EV figures and tables), but without the figures included. Please make sure that changes are highlighted to be clearly visible. Figure legends should be compiled at the end of the manuscript text.
- 2) individual production quality figure files as .eps, .tif, .jpg (one file per figure), of main figures and EV figures. Please upload these as separate, individual files upon re-submission.

The Expanded View format, which will be displayed in the main HTML of the paper in a collapsible format, has replaced the Supplementary information. You can submit up to 5 images as Expanded View. Please follow the nomenclature Figure EV1, Figure EV2 etc. The figure legend for these

should be included in the main manuscript document file in a section called Expanded View Figure Legends after the main Figure Legends section. Additional Supplementary material should be supplied as a single pdf file labeled Appendix. The Appendix should have page numbers and needs to include a table of content on the first page (with page numbers) and legends for all content. Please follow the nomenclature Appendix Figure Sx, Appendix Table Sx etc. throughout the text, and also label the figures and tables according to this nomenclature.

For more details please refer to our guide to authors:

See also our guide for figure preparation:

http://wol-prod-cdn.literatumonline.com/pb-assets/embosite/EMBOPress_Figure_Guidelines_061115-1561436025777.pdf

4) a complete author checklist, which you can download from our author guidelines (<https://www.embopress.org/page/journal/14693178/authorguide>). Please insert page numbers in the checklist to indicate where the requested information can be found in the manuscript. The completed author checklist will also be part of the RPF.

Please also follow our guidelines for the use of living organisms, and the respective reporting guidelines: <http://www.embopress.org/page/journal/14693178/authorguide#livingorganisms>

5) that primary datasets produced in this study (e.g. RNA-seq, ChIP-seq and array data) are deposited in an appropriate public database. This is now mandatory (like the COI statement). If no primary datasets have been deposited in any database, please state this in this section (e.g. 'No primary datasets have been generated and deposited').

The accession numbers and database should be listed in a formal "Data Availability " section (placed after Materials & Methods) that follows the model below. Please note that the Data Availability Section is restricted to new primary data that are part of this study.

Data availability

6) We strongly encourage the publication of original source data with the aim of making primary data more accessible and transparent to the reader. The source data will be published in a separate source data file online along with the accepted manuscript and will be linked to the relevant figure. If you would like to use this opportunity, please submit the source data (for example scans of entire gels or blots, data points of graphs in an excel sheet, additional images, etc.) of your key experiments together with the revised manuscript. If you want to provide source data, please include size markers for scans of entire gels, label the scans with figure and panel number, and send one PDF file per figure.

8) Regarding data quantification and statistics, please specify, where applicable, the number "n" for how many independent experiments (biological or technical replicates) were performed, the bars and error bars (e.g. SEM, SD) and the test used to calculate p-values in the respective figure legends. Please provide statistical testing where applicable, and also add a paragraph detailing this to the methods section. See: <http://www.embopress.org/page/journal/14693178/authorguide#statisticalanalysis>

9) Please add up to 5 key words to the title page.

10) Please also note our new reference format: <http://www.embopress.org/page/journal/14693178/authorguide#referencesformat>

I look forward to seeing a revised version of your manuscript when it is ready. Please let me know if you have questions or comments regarding the revision.

Kind regards,

Achim

Achim Breiling
Editor
EMBO Reports

Referee #1:

Comments on Novelty/Model System for Author:

The statistical analysis is clearly described and justified including the numbers of mice and repeats.

Novelty - It is not clear the novelty, regarding the efficacy of the cGAMP-VLP compared to other encapsulated cGAMP vaccines such as nano- or microparticle of cGAMP vaccines in the literature.

Moderate CD8+ T cells response and no protection by cGAMP - VLP in the VV-HIV-Gag model (FigS2) lowered the enthusiasm and impact of the study.

The authors used multiple models for different infections including a live influenza infection.

Remarks for Author:

In the manuscript titled "cGAMP loading enhances the immunogenicity of VLP vaccines" by Chauveau L et al., the authors described a new cGAMP-VLP vaccine platform that can be adopted to generate antibody and Th1 responses. In all, the data are solid and justifies the conclusion. The novelty as well as the limitation of the current study in comparison to other nano-, microparticle cGAMP vaccines, however, needs to be clearly stated. The data presentation and organization also need improvement. Below are my detailed comments.

Major Points:

1. The CD8 T cells response were not strong and there was a lack of protection in VV-HIV model. The authors may want to try a boost to enhance these responses that are essential for protections against viral infections.
2. The authors need to examine other vaccine-induced memory CD4 T cells responses, i.e. Th2 and Th17, in the cGAMP-VLP vaccine platform.
3. Need to discuss the limitation of the cGAMP-VLP platform. For example, vaccines are for the general public. How stable is cGAMP-VLP? Does it need special condition for storage and transportation? How does the cGAMP-VLP platform compares to the nano and microparticle cGAMP vaccines?

Minor Points:

1. In the title, please spell out "VLP"
2. In the Abstract, please clearly state the scientific question that this manuscript is addressing. "Based on these observations, we delivered cGAMP by inclusion within viral vaccine vectors" is not a scientific question.
3. The Introduction is unfocused and too long. Please focus on the literatures that directly related to the current study (cGAMP, cyclic dinucleotides-related vaccine), their current status and the knowledge/technology gap in this field.
4. The Discussion needs to address the future direction and potential problems of this cGAMP-VLP technology. For example, how do you improve its CD8 T cell responses. Antibody-mediated enhancement is a major issue in SARS vaccine development. cGAMP-VLP generate antibodies including antibodies to Gag, GFP, VSV and total VLP. Potential risk of antibody-mediated enhancement by cGAMP-VLP need to be discussed.
5. In many cases, Empty-VLP, without cGAMP, also generated vaccine responses, please discuss the mechanism and potential impact to the VLP vaccine platform.
6. Fonts in many figures are too small. Some are illegible (Figure 5A and 6C). I will suggest the authors to move some figure panels (e.g gating strategy for flow) to the Supplementary Materials and make the remaining panel big.

Referee #2:

The manuscript reports the generation of HIV-Gag-based VLPs that are produced by 293T cells that were transfected with a wt or inactive cGAMP expression vector. This results in VLPs with or without detectable cGAMP levels. A single immunization of C57/Bl6 mice with cGAMP-VLPs results in higher percentages of splenic CD4 T cells that express IL-2, IFN-g and TNF as compared to empty VLP immunized mice, when T splenocytes were restimulated with cGAMP-VLPs. Immunization with cGAMP-VLPs also resulted in higher percentages of CD8+T cells directed against an HIV-Gag-derived CD8+ T cell epitope. Serum IgG2a/c and -2b titers against cGAMP-VLPs and VSV-G are significantly higher in cGAMP-VLP than those observed in empty VLP immunized mice. This difference in humoral responses is lost in STING-deficient mice. Immunization with cGAMP-VLPs induced higher frequencies of Tfh cells in draining lymph nodes compared to empty VLPs. Increased VLP lysate reactive ASCs were also higher in cGAMP-VLP immunized mice. Immunization with cGAMP-VLPs that display influenza PR8 HA induced higher titers of microneutralizing serum abs than empty VLPs and protected better (at a lower immunogen dose, still protection) against PR8 virus challenge. Immunization with HA-displaying cGAMP-VLPs or Addavax adjuvanted VLPs protected equally well against PR8 virus challenge and induced comparable levels of neutralizing antibodies. Immunization with SARS-CoV-2 spike-displaying cGAMP-VLPs or empty VLPs induced non-detectable neutralizing antibodies in some mice and comparable levels of neutralizing antibodies in other mice. The paper is well written and experiments are well performed. cGAMP has been proposed as an adjuvant in several published studies before. It has been reported that inclusion of cGAMP in dextran particles improves its adjuvant activity considerably, e.g. when recombinant influenza hemagglutinin is adsorbed to such particles (Junkins et al., JCR, 2018).

Major remarks:

1. Line 139: 70 ng of cGAMP is associated with 1 million IUs of VLPs. Assuming that the MW of 1 VLP approximates 500 million Dalton, this would correspond to a mass of about 0.8 ng for 1 million VLPs. Is a 70:0.8 mass ratio of cGAMP:VLPs realistic? Same question for the molar ratio? The ELISA should be complemented with a bio-assay to estimate the amount of cGAMP that is incorporated in the VLPs.
2. The T cell responses were monitored in cGAMP-VLP stimulated immune cells (Fig. 2A-C). In Fig 3A antibody titers against lysates of cGAMP-VLPs are monitored. In other words, responses directed against the homologous immunogen were monitored in these cases for the cGAMP-VLP recipients. The reciprocal stimulation should be included in Fig. 2A-C and Fig. 3A. It is unclear whether the VLP lysates used in the ASC ELISPOT shown in Fig. 4F were prepared from empty or cGAMP-VLPs. If cGAMP-VLP lysates were used, it is also important to demonstrate that ASCs against lysates of empty VLPs are also more numerous in the cGAMP-VLP than in the empty VLP immunized groups.
3. The authors should document that the amount of conformationally intact HA is comparable in empty and cGAMP-VLPs. This could be shown by comparing the hemagglutination activity of the VLPs or with the H1/H5 HA-specific human mAb 21D85A.
4. Given that the VLPs lack incorporated nucleic acids, why does it take 24h before cells that have been incubated with the GAG-GFP + HA VLPs become GFP and HA positive?
5. The clinical potential of the approach could be documented better. Which production system could gain approval for clinical use? Other questions that should be addressed in the discussion are the scalability and yield of the production of cGAMP-VLPs. Vaccine recipients would likely seroconvert to HIV-Gag as well. How stable are the VLPs over time at RT and 4 degrees?

Other remarks:

1. Line 264: protection against influenza correlates with serum antibodies that have hemagglutination inhibition activity. Please correct the statement.
2. Lines 269, 271 and elsewhere in the text: "infection with ... VLPs". The VLPs lack nucleic acids and cannot replicate so "infection" seems incorrect to describe the transfer of material into cells by VLP take up.

Referee #3:

Based on preclinical evidence that agonist ligands for the stimulator of interferon genes (STING) receptor can be potent vaccine adjuvants, Chauveau et al. explored the hypothesis that inclusion of the STING ligand cGAMP in viral vaccine vectors protects cGAMP from degradation in the extracellular environment and enhances immunogenicity in mice.

The authors show that cGAMP can be efficiently packaged into HIV-derived virus-like particles (VLPs), enhancing polyfunctional CD4 and CD8 T cell responses to VLP antigens. Moreover, they observed that immunisation with cGAMP-VLPs induced production of virus neutralising antibodies, by enhancing the accumulation of Tfh cells in draining lymph nodes and promoting the development of antibody-secreting cells.

Following these observations, the authors explored whether cGAMP-VLPs incorporating IAV HA could confer protection following a live virus challenge. Their results show that vaccination with VLPs incorporating IAV HA induced neutralising antibodies and protected mice against subsequent IAV challenge.

To be timely, the authors produced also VLPs expressing SARS-CoV-2 Spike protein, to observe a mild increase in SARS-CoV-2 neutralising antibody titers upon immunisation with cGAMP vectors. Based on their results, the authors conclude by underlining the versatility and the potential of cGAMP-loaded VLPs as vaccine vectors.

The paper is well written and well presented. The data are convincing and the experiments well performed and complete.

The study has nonetheless the limitation of not being particularly novel, as the efficacy of STING ligands as potential adjuvants to induce T-cell and antibody responses has been previously shown by several groups (cf. References 16-18, 23-36 in the paper).

The main novelty of the paper lies in the incorporation of cGAMP into VLPs, allowing for a reduction in the dose of the adjuvant needed to induce protective immunity.

Minor comments:

1. The authors should discuss or reformulate the paragraph line 422 to 425 which is partly inconsistent with the findings of the second paragraph of results page 8, and in particular with the affirmation of the title (lines 148, 149) and the summary of the paragraph (lines 191, 192).
2. Supplementary Figure 7 does not exist (lines 338, 340, 341), it is supplementary Figure 5; the authors should correct them.

Response to referee #1Comments on Novelty/Model System for Author:

The statistical analysis is clearly described and justified including the numbers of mice and repeats.

We thank the reviewer for his/her positive feedback.

Novelty - It is not clear the novelty, regarding the efficacy of the cGAMP-VLP compared to other capsulated cGAMP vaccines such as nano- or microparticle of cGAMP vaccines in the literature.

We added a paragraph to the discussion to compare our strategy with vaccines incorporating cGAMP in micro- or nanoparticles (lines 404-411). We conclude that our study is the first to describe a strategy in which the vaccine associates both antigens and very low levels of cGAMP in the same entity and is still efficient after only one immunisation. Please also see our response to all reviewers above.

Moderate CD8+ T cells response and no protection by cGAMP - VLP in the VV-HIV-Gag model (FigS2) lowered the enthusiasm and impact of the study.

We discuss these results in lines 462-472 and propose prime-boost strategies for future studies to improve the CD8 T cell response. In this manuscript, we chose to focus on the strong antibody response because it was induced after just one dose of our vaccine and because most licensed vaccines confer protection via antibodies.

The authors used multiple models for different infections including a live influenza infection.

We thank the reviewer for highlighting that we used multiple models.

Remarks for Author:

In the manuscript titled "cGAMP loading enhances the immunogenicity of VLP vaccines" by Chauveau L et al., the authors described a new cGAMP-VLP vaccine platform that can be adopted to generate antibody and Th1 responses. In all, the data are solid and justifies the conclusion. The novelty as well as the limitation of the current study in comparison to other nano-, microparticle cGAMP vaccines, however, needs to be clearly stated. The data presentation and organization also need improvement. Below are my detailed comments.

We are grateful that the reviewer assessed our data as "solid" and "justifying the conclusion". We have now described better how our strategy compares to other nanoparticle cGAMP vaccines and hope this comparison highlights the advantages of our approach (lines 404-411).

Major Points:

1. The CD8 T cells response were not strong and there was a lack of protection in VV-HIV model. The authors may want to try a boost to enhance these responses that are essential for protections against viral infections.

We thank the reviewer for this suggestion, which is included in the discussion (lines 462-472). As detailed above, we chose to focus this study on the antibody response induced by our vaccine platform.

2. The authors need to examine other vaccine-induced memory CD4 T cells responses, i.e. Th2 and Th17, in the cGAMP-VLP vaccine platform.

We observed a Th1-type response to our vaccine with induction of CD4 T cells secreting TNF α and IFN γ as well as antibody responses skewed towards IgG2a/c and IgG2b isotypes. The

lack of induction of IgG1 antibodies suggests that Th2 responses were not induced. Indeed, the reviewer highlights above that we describe Th1 responses. We believe that the question whether small Th2 and Th17 responses are induced is a mechanistic detail. Given the editorial guidance that “*EMBO reports* emphasizes novel functional over detailed mechanistic insight”, we have not pursued such experiments.

3. Need to discuss the limitation of the cGAMP-VLP platform. For example, vaccines are for the general public. How stable is cGAMP-VLP? Does it need special condition for storage and transportation? How does the cGAMP-VLP platform compares to the nano and microparticle cGAMP vaccines?

Our manuscript is a proof-of-concept study describing the adaptive immune response induced by our new cGAMP-VLP vaccine platform. Our objective was to demonstrate that inclusion of cGAMP in a viral-vectored vaccine is an efficient and convenient strategy for the design of new and safe vaccines. However, the cGAMP-VLPs described here are not ready for use “as-is” and will require up-scaling and further development before they can be used in the clinic. In order to keep the discussion concise, we therefore do not believe it is necessary to discuss the details of manufacturing, storing or transporting cGAMP-VLPs in our manuscript.

Minor Points:

1. In the title, please spell out "VLP"

Done.

2. In the Abstract, please clearly state the scientific question that this manuscript is addressing. "Based on these observations, we delivered cGAMP by inclusion within viral vaccine vectors" is not a scientific question.

We have modified the abstract accordingly (lines 32-33): “Here, we investigated whether inclusion of cGAMP within viral vaccine vectors enhances their immunogenicity.”

3. The Introduction is unfocused and too long. Please focus on the literatures that directly related to the current study (cGAMP , cyclic dinucleotides-related vaccine), their current status and the knowledge/technology gap in this field.

Our paper is likely to be of interest to a broad readership including immunologists, virologists and the vaccine community. It is therefore important to introduce the key immune responses we are characterising in addition to the literature directly related to the cGAMP-VLPs as vaccines. As per the reviewer’s request, we shortened the introduction where possible. We believe that this is now a good introduction for the audience.

4. The Discussion needs to address the future direction and potential problems of this cGAMP-VLP technology. For example, how do you improve its CD8 T cell responses. Antibody-mediated enhancement is a major issue in SARS vaccine development. cGAMP-VLP generate antibodies including antibodies to Gag, GFP, VSV and total VLP. Potential risk of antibody-mediated enhancement by cGAMP-VLP need to be discussed.

We discuss how CD8 T cell responses may be improved by using a prime-boost strategy (lines 462-472). Recent clinical trials of SARS-CoV-2 vaccines did not reveal antibody-mediated enhancement as a particular risk. We therefore do not believe it is necessary to include a discussion about potential risks of antibody-mediated enhancement.

5. In many cases, Empty-VLP, without cGAMP, also generated vaccine responses, please discuss the mechanism and potential impact to the VLP vaccine platform.

The reviewer is correct; Empty-VLPs were not inert but induced adaptive immune responses, albeit at lower levels than cGAMP-VLPs. This effect may be due to the presence in VLP preparations of nucleic acid fragments from producer cells, which could mediate a degree of

adjuvanticity via receptors such as TLR9. We have added this idea to the discussion (lines 431-434) and cited a relevant reference (Pichlmair *et al*, 2007).

6. Fonts in many figures are too small. Some are illegible (Figure 5A and 6C). I will suggest the authors to move some figure panels (e.g gating strategy for flow) to the Supplementary Materials and make the remaining panel big.

We have increased font sizes in these figures as requested by the reviewer but prefer to show some of the raw data (i.e. FACS gating) in the main manuscript for transparency.

Response to reviewer #2

Remarks for Author:

The manuscript reports the generation of HIV-Gag-based VLPs that are produced by 293T cells that were transfected with a wt or inactive cGAS expression vector. This results in VLPs with or without detectable cGAMP levels. A single immunization of C57/Bl6 mice with cGAMP-VLPs results in higher percentages of splenic CD4 T cells that express IL-2, IFN- γ and TNF as compared to empty VLP immunized mice, when T splenocytes were restimulated with cGAMP-VLPs. Immunization with cGAMP-VLPs also resulted in higher percentages of CD8+T cells directed against an HIV-Gag-derived CD8+ T cell epitope. Serum IgG2a/c and -2b titers against cGAMP-VLPs and VSV-G are significantly higher in cGAMP-VLP than those observed in empty VLP immunized mice. This difference in humoral responses is lost in STING-deficient mice. Immunization with cGAMP-VLPs induced higher frequencies of Tfh cells in draining lymph nodes compared to empty VLPs. Increased VLP lysate reactive ASCs were also higher in cGAMP-VLP immunized mice.

Immunization with cGAMP-VLPs that display influenza PR8 HA induced higher titers of microneutralizing serum abs than empty VLPs and protected better (at a lower immunogen dose, still protection) against PR8 virus challenge. Immunization with HA-displaying cGAMP-VLPs or Addavax adjuvanted VLPs protected equally well against PR8 virus challenge and induced comparable levels of neutralizing antibodies. Immunization with SARS-CoV-2 spike-displaying cGAMP-VLPs or empty VLPs induced non-detectable neutralizing antibodies in some mice and comparable levels of neutralizing antibodies in other mice.

De paper is well written and experiments are well performed. cGAMP has been proposed as an adjuvant in several published studies before. It has been reported that inclusion of cGAMP in dextran particles improves its adjuvant activity considerably, e.g. when recombinant influenza hemagglutinin is adsorbed to such particles (Junkins et al., JCR, 2018).

We thank the reviewer for his/her summary and for highlighting that our manuscript is well written and that experiments were well done.

The novelty of our study is not in proposing to use cGAMP as an adjuvant (which has been published before as the reviewer points out) but rather in associating the adjuvant cGAMP with antigens in the same entity, i.e. the VLPs. This allowed us to increase efficiency of the vaccine by protecting cGAMP from the extracellular milieu and by delivering it to the same cells that also receive the antigens. As we discuss in the manuscript, this allowed us to use amounts of cGAMP that are lower than any other study using it as an adjuvant, including the studies where it is encapsulated in micro- or nanoparticles. We have now added a more detailed discussion of cGAMP-VLPs in comparison with cGAMP containing micro- and nanoparticles (lines 404-411).

Major remarks:

1. Line 139: 70 ng of cGAMP is associated with 1 million IUs of VLPs. Assuming that the MW of 1 VLP approximates 500 million Dalton, this would correspond to a mass of about 0.8 ng for 1 million VLPs. Is a 70:0.8 mass ratio of cGAMP:VLPs realistic? Same question for the molar ratio? The ELISA should be complemented with a bio-assay to estimate the amount of cGAMP that is incorporated in the VLPs.

Bioassays are inherently more variable than ELISAs, which had led us to choose ELISA as a method to quantify cGAMP in the initial submission. However, as requested, we now performed a cGAMP bioassay using a protocol previously published by us (Bridgeman *et al*, 2015). As shown in the new Figure 1B, the levels of cGAMP detected were broadly within the same range as the ELISA data but with a tendency to be somewhat higher.

Please note that, while going through the calculations of the cGAMP ELISA results again, we uncovered a small calculation mistake for some VLP batches. This has been rectified in the

new Figure 1B; the VLP batches used later in the manuscript for comparing cGAMP inside and outside of VLPs were not affected.

We also measured cGAMP in cGAMP-VLPs by mass spectrometry in collaboration with the laboratory of V. Kaefer (Hannover, Germany). These experiments also showed results within the same range but were not included as they were performed with VLP batches not used in this study. With three different read-outs generating similar results we are confident in our data. Importantly, no cGAMP was found in Empty-VLPs by all three methods.

Finally, VSV-G pseudotyped lentiviral preparations are well known to contain high levels of enveloped vesicles, which are present in much higher numbers than infectious particles (Bess *et al*, 1997; Gluschankof *et al*, 1997; Pichlmair *et al*, 2007). Given the enveloped nature of these vesicles, it is likely that they contain cGAMP. Therefore, the actual cGAMP : (infectious) VLP mass and molar ratios are likely much lower than estimated by the reviewer and are impossible to calculate with precision.

2. The T cell responses were monitored in cGAMP-VLP stimulated immune cells (Fig. 2A-C). In Fig 3A antibody titers against lysates of cGAMP-VLPs are monitored. In other words, responses directed against the homologous immunogen were monitored in these cases for the cGAMP-VLP recipients. The reciprocal stimulation should be included in Fig. 2A-C and Fig. 3A. It is unclear whether the VLP lysates used in the ASC ELISPOT shown in Fig. 4F were prepared from empty or cGAMP-VLPs. If cGAMP-VLP lysates were used, it is also important to demonstrate that ASCs against lysates of empty VLPs are also more numerous in the cGAMP-VLP than in the empty VLP immunized groups.

We thank the reviewer for highlighting this technical caveat of our work. We added new data to the revised manuscript that address these points. We restimulated spleen cells with Empty-VLP-loaded DCs instead of cGAMP-VLPs. In this setting, we observed the same increase in CD4 T cell responses in splenocytes from mice immunised with cGAMP-VLPs compared to Empty-VLPs. We have now added this result in Fig EV1 (new panels A and B). Moreover, we repeated the IgG2b ELISA shown in Fig 3A using Empty-VLP lysates for coating of the ELISA plates (new Fig EV3C). The results were very similar to those obtained before using cGAMP-VLP lysates for coating. We are thus confident that the antibody responses we report were not directed against the cGAMP contained in cGAMP-VLPs. Based on this conclusion, we have not repeated the experiment in Fig 4F (which was done with cGAMP-VLP lysates; now stated in line 1113) with Empty-VLP lysates.

3. The authors should document that the amount of conformationally intact HA is comparable in empty and cGAMP-VLPs. This could be shown by comparing the hemagglutination activity of the VLPs or with the H1/H5 HA-specific human mAb 21D85A.

In our flow cytometry analysis in Fig EV4B, we already used the H1/H5 HA-specific human mAb 21D85A suggested by the reviewer. This experiment therefore showed that the HA transferred to target cells was conformationally intact and was transferred in comparable amounts by Empty- and cGAMP-HA-VLPs.

4. Given that the VLPs lack incorporated nucleic acids, why does it take 24h before cells that have been incubated with the GAG-GFP + HA VLPs become GFP and HA positive?

We apologise that this was not clear. The time point for looking at infectivity was chosen because in the same experiment we measured IFN produced; therefore, a 24 hour time point was more suitable. However, as shown in the figure below, we could detect GFP in cells incubated with cGAMP-VLPs and Empty-VLPs at 4 hours post-infection. The GFP signal was slightly higher at 4 hours compared to 24 hours, which was expected as GFP is transferred only as protein and will degrade over time.

HEK293 cells were infected with decreasing amounts of cGAMP-VLPs and Empty-VLPs as described for the infectivity assay in Figure 1 of the manuscript. The infection was monitored at 4 hours and 24 hours post-infection by quantifying GFP+ cells by flow cytometry. Data are from one VLP batch tested in technical duplicates; mean and range are shown.

5. The clinical potential of the approach could be documented better. Which production system could gain approval for clinical use? Other questions that should be addressed in the discussion are the scalability and yield of the production of cGAMP-VLPs. Vaccine recipients would likely seroconvert to HIV-Gag as well. How stable are the VLPs over time at RT and 4 degrees?

As detailed in our response to reviewer 1 (major point 3), this is a proof-of-concept study and – in order to keep the discussion concise – we do not believe it is necessary to discuss the details of VLP production or stability at this point in the development of the vaccine strategy.

Other remarks:

1. Line 264: protection against influenza correlates with serum antibodies that have hemagglutination inhibition activity. Please correct the statement.

Done.

2. Lines 269, 271 and elsewhere in the text: "infection with ... VLPs". The VLPs lack nucleic acids and cannot replicate so "infection" seems incorrect to describe the transfer of material into cells by VLP take up.

Semantically, we agree with the reviewer that 'infection' has a different meaning for VLPs compared to viruses. However, alternative wording would be convoluted; we have therefore decided to continue using 'infection'.

Response to reviewer #3

Remarks for Author:

Based on preclinical evidence that agonist ligands for the stimulator of interferon genes (STING) receptor can be potent vaccine adjuvants, Chauveau et al. explored the hypothesis that inclusion of the STING ligand cGAMP in viral vaccine vectors protects cGAMP from degradation in the extracellular environment and enhances immunogenicity in mice.

The authors show that cGAMP can be efficiently packaged into HIV-derived virus-like particles (VLPs), enhancing polyfunctional CD4 and CD8 T cell responses to VLP antigens. Moreover, they observed that immunisation with cGAMP-VLPs induced production of virus neutralising antibodies, by enhancing the accumulation of Tfh cells in draining lymph nodes and promoting the development of antibody-secreting cells.

Following these observations, the authors explored whether cGAMP-VLPs incorporating IAV HA could confer protection following a live virus challenge. Their results show that vaccination with VLPs incorporating IAV HA induced neutralising antibodies and protected mice against subsequent IAV challenge.

To be timely, the authors produced also VLPs expressing SARS-CoV-2 Spike protein, to observe a mild increase in SARS-CoV-2 neutralising antibody titers upon immunisation with cGAMP vectors.

Based on their results, the authors conclude by underlining the versatility and the potential of cGAMP-loaded VLPs as vaccine vectors.

The paper is well written and well presented. The data are convincing and the experiments well performed and complete.

The study has nonetheless the limitation of not being particularly novel, as the efficacy of STING ligands as potential adjuvants to induce T-cell and antibody responses has been previously shown by several groups (cf. References 16-18, 23-36 in the paper).

The main novelty of the paper lies in the incorporation of cGAMP into VLPs, allowing for a reduction in the dose of the adjuvant needed to induce protective immunity.

We thank the reviewer for this nice summary of our manuscript and for acknowledging that our data is “convincing” and that the experiments are “well performed and complete”. The reviewer also highlights that the goal of our paper was not to show cGAMP is an adjuvant (which is already known) but rather to describe and show the efficiency of a vaccine strategy where cGAMP is incorporated in a virus particle vaccine, as discussed above.

Minor comments:

1. The authors should discuss or reformulate the paragraph line 422 to 425 which is partly inconsistent with the findings of the second paragraph of results page 8, and in particular with the affirmation of the title (lines 148, 149) and the summary of the paragraph (lines 191, 192). Thank you for pointing this out; we have rephrased the discussion (lines 463-464 in the revised manuscript).

2. Supplementary Figure 7 does not exist (lines 338, 340, 341), it is supplementary Figure 5; the authors should correct them.

Thank you for spotting this error; this has been corrected.

References

Bess JW, Jr., Gorelick RJ, Bosche WJ, Henderson LE, Arthur LO (1997) Microvesicles are a source of contaminating cellular proteins found in purified HIV-1 preparations. *Virology* 230: 134-144

Bridgeman A, Maelfait J, Davenne T, Partridge T, Peng Y, Mayer A, Dong T, Kaever V, Borrow P, Rehwinkel J (2015) Viruses transfer the antiviral second messenger cGAMP between cells. *Science* 349: 1228-1232

Gluschankof P, Mondor I, Gelderblom HR, Sattentau QJ (1997) Cell membrane vesicles are a major contaminant of gradient-enriched human immunodeficiency virus type-1 preparations. *Virology* 230: 125-133

Pichlmair A, Diebold SS, Gschmeissner S, Takeuchi Y, Ikeda Y, Collins MK, Reis e Sousa C (2007) Tubulovesicular structures within vesicular stomatitis virus G protein-pseudotyped lentiviral vector preparations carry DNA and stimulate antiviral responses via Toll-like receptor 9. *J Virol* 81: 539-547

Dear Prof. Rehwinkel,

Thank you for the submission of your revised manuscript to our editorial offices. We have now received the reports from the three referees that were asked to re-evaluate your study, you will find below. As you will see, the referee support publication of the study in EMBO reports. Nevertheless, referees #1 and #2 have remaining concerns and suggestions to improve the manuscript, we ask you to address in a final revised version. Please also provide a point-by-point response regarding these remaining points.

Moreover, I have these editorial requests I also ask you to address:

- Please provide the abstract written in present tense.
- Please call the 'Declaration of interests' 'Conflict of interest statement'.
- There seem to be no callouts for Fig EV2C and Fig EV5B. Please check.
- For Fig. EV4B/C you indicate that the data shown comes from 2 replicates. In that case, please show these data without statistics, by showing the two datasets separated. This is much more transparent and illustrates better the data.
- Could statistical testing be done for the data shown in Figs. 5B and 6C? Please make sure that the number "n" for how many independent experiments were performed, their nature (biological versus technical replicates), the bars and error bars (e.g. SEM, SD) and the test used to calculate p-values is indicated in the respective figure legends (also of the EV figures), and that statistical testing has been done where applicable.
- Please add to the data availability statement a sentence that no primary datasets have been deposited in any database (e.g. 'No primary datasets have been generated or deposited').
- In the references, please add for Huang et al. the DOI and mark this reference with [PREPRINT]. See also: <https://www.embopress.org/page/journal/14693178/authorguide#referencesformat>
- Please upload the information in the Appendix as 'Reagents and Tools table'. I have attached templates for that in word or excel format. Please upload the filled in table to the manuscript tracking system as a 'Reagent Table' file. Please add the Appendix references to the main references. This example shows how the table will display in the published article and includes examples of the type of information that should be provided for the different categories of reagents and tools. Please list your reagents/tools using the categories provided in the template and do not add additional subheadings to the table. Reagents/tools that do not fit in any of the specific categories can be listed under "Other":
https://www.embopress.org/pb%2Dassets/embo-site/msb_177951_sample_FINAL.pdf
- Finally, please find attached a word file of the manuscript text (provided by our publisher) with changes we ask you to include in your final manuscript text, and some queries, we ask you to address. Please provide your final manuscript file with track changes, in order that we can see any modifications done.

In addition, I would need from you:

- a short, two-sentence summary of the manuscript (around 35 words).
- three to four short bullet points highlighting the key findings of your study
- a schematic summary figure (in jpeg or tiff format with the exact width of 550 pixels and a height of not more than 400 pixels) that can be used as a visual synopsis on our website.

Kind regards,

Achim Breiling
Editor
EMBO Reports

Referee #1:

Unfortunately, two of my major points were not addressed adequately in the Revision.

Major Point 2: The authors need to examine other vaccine-induced memory CD4 T cells responses, i.e. Th2 and Th17, in the cGAMP-VLP vaccine platform.

Previous studies have established that CDNs generate Th1/Th2/Th17 and CD8 T cells vaccine responses. Such balanced adjuvanticity makes CDNs a superior adjuvant to Th1-biased or Th2-biased adjuvants. In Figure 3A, VLP generates IgG1 response indicating a Th2 response.

Nevertheless, in the authors' response, they did not provide ICS stain of IL4+ CD4 T cells and IL-17+ CD4 T cells. Thus the characterization of the VLP platform is incomplete. Is cGAMP-VLP a Th1-biased adjuvant that only suitable for viral and intracellular pathogens or it also induces Th2, Th17 adjuvant responses that may have a broader application.

Major Point 3. Need to discuss the limitation of the CGAMP-VLP platform. For example, vaccines are for the general public. How stable is cGAMP-VLP? Does it need special conditions for storage and transportation?

The authors could address these issues in the Discussion as future research directions rather than ignore it. Furthermore, safety is a top priority for vaccine development. VLP is made in human cells (HEK 293 cells). Is it possible that VLP may contain human antigen(s)? If so, when used as vaccines, these human cell-derived VLP may generate antibodies against self-human antigens that could result in autoimmune disease.

Minor Point:

1. In the revision, the authors used a second method (BioAssay) to determine VLP cGAMP concentration. The authors stated the result was 2~3 fold higher than that of ELISA (Figure 1B) (Line 152-154). To avoid confusion, the authors may want to a) state which method is more accurate in reflecting cGAMP concentration in VLP? b) explain the discrepancy.

Referee #2:

The authors report on an elegant system to incorporate cGAMP in enveloped VLPs that can be pseudotyped with VSV-G, influenza hemagglutinin or SARS-CoV-2 fusion proteins. Immunization of B16 mice by intramuscular injection of these VLPs, and much less so of VLPs without incorporated cGAMP, can induce antigen-specific Th1 skewed T cell responses and neutralizing antibody responses against viruses with the corresponding enveloped proteins.

Compared to an earlier version of the manuscript, results of a cGAMP bioassay to quantify VLP-incorporated cGAMP levels, as well as immune serum reactivity against coated lysates of empty VLPs, were added. I greatly appreciate the inclusion of the results of these additional experiments. The presented work is of a high technical quality. The manuscript is very well written.

I have one remaining comment:

I did not find or overlooked reference to the use of mAb 21-D8-5A in the manuscript, except for its mentioning in the appendix. The authors may want to include this information in the materials and methods section.

Referee #3:

The authors performed suitable changes. As mentioned in my earlier report, the experiments are well performed and the paper is clear.

It confirms and reinforces the key concept of the efficacy of STING ligands as potential adjuvants to induce immune responses as shown in previous studies.

The paper demonstrates here the interest of using cGAMP in VLPs to boost vaccine immunogenicity, which is fair.

Point-by-point reply

EMBOR-2021-52447-V2

Inclusion of cGAMP within virus-like particle vaccines enhances their immunogenicity

Response to all reviewers

The authors would like to thank all reviewers for carefully examining our revised manuscript. We have made further revisions in text and hope our manuscript is now ready for publication. New or revised text is shown purple in the revised manuscript. Red indicates changes made during the previous round of revision.

Response to editorial requests

- Please provide the abstract written in present tense.
- Please call the 'Declaration of interests' 'Conflict of interest statement'.

This has been changed as requested.

- There seem to be no callouts for Fig EV2C and Fig EV5B. Please check.

Callouts for these figures were included.

- For Fig. EV4B/C you indicate that the data shown comes from 2 replicates. In that case, please show these data without statistics, by showing the two datasets separated. This is much more transparent and illustrates better the data.

We included data from a third repeat.

- Could statistical testing be done for the data shown in Figs. 5B and 6C? Please make sure that the number "n" for how many independent experiments were performed, their nature (biological versus technical replicates), the bars and error bars (e.g. SEM, SD) and the test used to calculate p-values is indicated in the respective figure legends (also of the EV figures), and that statistical testing has been done where applicable.

We included statistical testing as requested and double-checked all data information provided.

- Please add to the data availability statement a sentence that no primary datasets have been deposited in any database (e.g. 'No primary datasets have been generated or deposited').

This was added as requested.

- In the references, please add for Huang et al. the DOI and mark this reference with [PREPRINT]. See also:
<https://www.embopress.org/page/journal/14693178/authorguide#referencesformat>

This manuscript is now published and we updated the reference.

- Please upload the information in the Appendix as 'Reagents and Tools table'. I have attached templates for that in word or excel format. Please upload the filled in table to the manuscript tracking system as a 'Reagent Table' file. Please add the Appendix references to the main references. This example shows how the table will display in the published article and includes examples of the type of information that should be provided for the different categories of reagents and tools. Please list your reagents/tools using the categories provided in the template and do not add additional subheadings to the table. Reagents/tools that do not fit in any of the specific categories can be listed under "Other":
https://www.embopress.org/pb%2Dassets/embo-site/msb_177951_sample_FINAL.pdf

We prepared this table as requested.

- Finally, please find attached a word file of the manuscript text (provided by our publisher) with changes we ask you to include in your final manuscript text, and some queries, we ask you to address. Please provide your final manuscript file with track changes, in order that we can see any modifications done.

We addressed all queries. Changes are highlighted in purple.

In addition, I would need from you:

- a short, two-sentence summary of the manuscript (around 35 words).
- three to four short bullet points highlighting the key findings of your study
- a schematic summary figure (in jpeg or tiff format with the exact width of 550 pixels and a height of not more than 400 pixels) that can be used as a visual synopsis on our website.

We have prepared these items and include them in the submission.

Response to referee #1

Unfortunately, two of my major points were not addressed adequately in the Revision.

We apologise for not addressing these points adequately in the previous revision and hope our additional changes answer this reviewer's questions.

Major Point 2: The authors need to examine other vaccine-induced memory CD4 T cells responses, i.e. Th2 and Th17, in the cGAMP-VLP vaccine platform. Previous studies have established that CDNs generate Th1/Th2/Th17 and CD8 T cells vaccine responses. Such balanced adjuvanticity makes CDNs a superior adjuvant to Th1-biased or Th2-biased adjuvants. In Figure 3A, VLP generates IgG1 response indicating a Th2 response. Nevertheless, in the authors' response, they did not provide ICS stain of IL4+ CD4 T cells and IL-17+ CD4 T cells. Thus the characterization of the VLP platform is incomplete. Is cGAMP-VLP a Th1-biased adjuvant that only suitable for viral and intracellular pathogens or it also induces Th2, Th17 adjuvant responses that may have a broader application.

While bacterial CDNs induce a balanced Th1/Th2/Th17 and CD8 T cell response, several studies have previously found that mammalian 2'3'-cGAMP induces mostly Th1 and weak Th2 responses. For example, a direct comparison of cyclic di-GMP and 2'3'-cGAMP showed similar induction of IgG2a but a better induction of IgG1 and IL17 in response to cyclic di-GMP (Blaauboer *et al*, 2015). This is in line with our results in Figure 3 showing that inclusion of cGAMP in VLPs increases IgG2a and IgG2b/c responses but does not significantly enhance the IgG1 response. This reviewer is right in saying that cGAMP-VLPs induce an IgG1 response (and thus probably Th2 responses), but this response is already induced in

Empty-VLP immunised mice. Our study focusses on the immune response induced by the inclusion of cGAMP in VLPs. Moreover, we are mostly interested in antiviral and thus Th1 responses. We therefore believe that further studies into Th2 and Th17 responses are beyond the scope of our manuscript and would unnecessarily delay publication as the required *in vivo* experiments are time-consuming. Whether the cGAMP-VLP platform allows simultaneous induction of Th1 responses by cGAMP and induction of Th2 responses by the VLP, which could be harnessed for broader applications, is an interesting question that should be investigated in future studies. We have therefore added a paragraph about this in the discussion (lines 457-462).

Major Point 3. Need to discuss the limitation of the CGAMP-VLP platform. For example, vaccines are for the general public. How stable is cGAMP-VLP? Does it need special conditions for storage and transportation? The authors could address these issues in the Discussion as future research directions rather than ignore it. Furthermore, safety is a top priority for vaccine development. VLP is made in human cells (HEK 293 cells). Is it possible that VLP may contain human antigen(s)? If so, when used as vaccines, these human cell-derived VLP may generate antibodies against self-human antigens that could result in autoimmune disease.

As pointed out in the previous round of revision, this study is a proof-of-concept, and further development of the cGAMP-VLP platform is needed to reach the quality and safety standards required for use in human. These questions are important but beyond the scope of our study. Nevertheless, we have now added some of these questions as future research directions in the discussion (lines 416-421). Of note, other viral vaccine vectors, such as the adenovirus-based vector used in the Oxford / AstraZeneca Covid-19 vaccine, are also produced in human cells, including HEK293 cells. As such, while the generation of self-antigens by such platforms is a legitimate concern, we believe it can be addressed in future development of the vaccine.

Minor Point:

1. In the revision, the authors used a second method (BioAssay) to determine VLP cGAMP concentration. The authors stated the result was 2~3 fold higher than that of ELISA (Figure 1B) (Line 152-154). To avoid confusion, the authors may want to a) state which method is more accurate in reflecting cGAMP concentration in VLP? b) explain the discrepancy.

It is difficult to be entirely sure which method is more accurate and to explain with certainty the discrepancy without performing in-depth comparisons of both methods. We believe that the ELISA, where cGAMP directly binds to antibodies coated on a plate, is likely more accurate than the bioassay requiring stimulation of THP-1 cells. We have added this reasoning to the Results section (lines 165-167) and chose to use the amount of cGAMP measured by ELISA for further experiments.

Response to referee #2

The authors report on an elegant system to incorporate cGAMP in enveloped VLPs that can be pseudotyped with VSV-G, influenza hemagglutinin or SARS-CoV-2 fusion proteins. Immunization of B16 mice by intramuscular injection of these VLPs, and much less so of VLPs without incorporated cGAMP, can induce antigen-specific Th1 skewed T cell responses and neutralizing antibody responses against viruses with the corresponding enveloped proteins. Compared to an earlier version of the manuscript, results of a cGAMP bioassay to quantify VLP-incorporated cGAMP levels, as well as immune serum reactivity against coated lysates

of empty VLPs, were added. I greatly appreciate the inclusion of the results of these additional experiments. The presented work is of a high technical quality. The manuscript is very well written.

We thank the reviewer for these nice comments on our revised manuscript.

I have one remaining comment:

I did not find or overlooked reference to the use of mAb 21-D8-5A in the manuscript, except for its mentioning in the appendix. The authors may want to include this information in the materials and methods section.

We now included the clone number of this antibody in the Results and Materials and Methods sections (lines 304 and 567, resp.), and provide a reference in the Reagents table.

Response to referee #3

The authors performed suitable changes. As mentioned in my earlier report, the experiments are well performed and the paper is clear.

It confirms and reinforces the key concept of the efficacy of STING ligands as potential adjuvants to induce immune responses as shown in previous studies.

The paper demonstrates here the interest of using cGAMP in VLPs to boost vaccine immunogenicity, which is fair.

We thank the reviewer for his/her positive assessment of our revised manuscript.

References

Blauboer SM, Mansouri S, Tucker HR, Wang HL, Gabrielle VD, Jin L (2015) The mucosal adjuvant cyclic di-GMP enhances antigen uptake and selectively activates pinocytosis-efficient cells in vivo. *Elife* 4

Prof. Jan Rehwinkel
University of Oxford
Radcliffe Department of Medicine
JR Hospital, WIMM, HIU
Headley Way
Oxford, Oxfordshire OX3 9DS
United Kingdom

Dear Prof. Rehwinkel,

I am very pleased to accept your manuscript for publication in the next available issue of EMBO reports. Thank you for your contribution to our journal.

At the end of this email I include important information about how to proceed. Please ensure that you take the time to read the information and complete and return the necessary forms to allow us to publish your manuscript as quickly as possible.

As part of the EMBO publication's Transparent Editorial Process, EMBO reports publishes online a Review Process File to accompany accepted manuscripts. As you are aware, this File will be published in conjunction with your paper and will include the referee reports, your point-by-point response and all pertinent correspondence relating to the manuscript.

If you do NOT want this File to be published, please inform the editorial office within 2 days, if you have not done so already, otherwise the File will be published by default [contact: emboreports@embo.org]. If you do opt out, the Review Process File link will point to the following statement: "No Review Process File is available with this article, as the authors have chosen not to make the review process public in this case."

Should you be planning a Press Release on your article, please get in contact with emboreports@wiley.com as early as possible, in order to coordinate publication and release dates.

Thank you again for your contribution to EMBO reports and congratulations on a successful publication. Please consider us again in the future for your most exciting work.

Yours sincerely,

Achim Breiling
Editor
EMBO Reports

THINGS TO DO NOW:

You will receive proofs by e-mail approximately 2-3 weeks after all relevant files have been sent to

our Production Office; you should return your corrections within 2 days of receiving the proofs.

Please inform us if there is likely to be any difficulty in reaching you at the above address at that time. Failure to meet our deadlines may result in a delay of publication, or publication without your corrections.

All further communications concerning your paper should quote reference number EMBOR-2021-52447V3 and be addressed to emboreports@wiley.com.

Should you be planning a Press Release on your article, please get in contact with emboreports@wiley.com as early as possible, in order to coordinate publication and release dates.

Corresponding Author Name: Jan Rehwinkel

Manuscript Number: EMBOR-2021-52447-T